# Shape and structural relaxation of colloidal tactoids

Hamed Almohammadi [1,5], Sayyed Ahmad Khadem[2,3,5], Massimo Bagnani [1], Alejandro D. Rey [2,3] &
Raffaele Mezzenga [1,4✉]

Facile geometric-structural response of liquid crystalline colloids to external fields enables many technological advances. However, the relaxation mechanisms for liquid crystalline colloids under mobile boundaries remain still unexplored. Here, by combining experiments, numerical simulations and theory, we describe the shape and structural relaxation of colloidal liquid crystalline micro-droplets, called tactoids, where amyloid fibrils and cellulose nano-crystals are used as model systems. We show that tactoids shape relaxation bears a universal single exponential decay signature and derive an analytic expression to predict this out of equilibrium process, which is governed by liquid crystalline anisotropic and isotropic contributions. The tactoids structural relaxation shows fundamentally different paths, with first- and second-order exponential decays, depending on the existence of splay/bend/twist orientation structures in the ground state. Our findings offer a comprehensive understanding on dynamic confinement effects in liquid crystalline colloidal systems and may set unexplored directions in the development of novel responsive materials.

[1] Department of Health Sciences and Technology, ETH Zurich, Zurich, Switzerland. [2] Department of Chemical Engineering, McGill University, Montreal, QC, Canada. [3] Quebec Centre for Advanced Materials, Canada (QCAM/CQMF), Montreal, QC, Canada. [4] Department of Materials, ETH Zurich, Zurich, Switzerland. [5] These authors contributed equally: Hamed Almohammadi, Sayyed Ahmad Khadem. ✉email: raffaele.mezzenga@hest.ethz.ch

Colloidal liquid crystals are a class of soft matter formed when shape-anisotropic nanoparticles are dispersed in an isotropic fluid[1]. When confined to a finite volume, rod-like colloidal particles self-organize into various structures that are set by a delicate balance between anisotropic viscoelastic and surface properties[2–4]. The subtle balance between these contributions results in facile response to external fields[1,5] such as mechanical, flow, electric, and magnetic, giving rise to many opportunities and technological applications. Examples include displays, spatial light modulator and tunable filters in medical devices and optics, liquid crystal biosensors for rapid diagnostics, and new functional material such as artificial muscles exploiting liquid crystalline anisotropic physical properties[6–8]. However, facile responsiveness to external fields (and disturbance) makes the colloidal liquid crystals very fragile to be studied experimentally under dynamical conditions[5,9]. In particular, relaxation of liquid crystalline droplets under mobile confinement is still poorly understood despite its central importance in a variety of phenomena in condensed matter physics. This includes particle packing[10], self-assembly[11], and relaxation of colloidal liquids[12] with implications in the field of active nematic, e.g., living liquid crystals[13], where the understanding of the hydrodynamics of the liquid crystals is critical[14].

Liquid crystalline droplets, known as tactoids, are a particularly significant example of colloidal liquid crystals, since they consist of micro-confined liquid crystalline colloids with a self-selected shape/structure stemming out from the thermodynamically-driven phenomena from which they emerge, i.e., spontaneous liquid–liquid crystalline phase separation[15–23]. In stark contrast with spherical liquid crystalline emulsions, achieved commonly by emulsifying liquid crystals in another immiscible liquid (like water in oil)[24,25], tactoids hold spindle-like, prolate, or oblate shapes with different nematic-cholesteric internal structures[2–4,15–22,26], as a consequence of the coupling between the vanishingly small interfacial tension, the surface anchoring at the interface, the chirality of colloids and the anisotropic elastic properties[3,4]. These features make tactoids a very unique system with peculiar viscoelastic[27–29] and boundary[3,4] properties, thus adding theoretical challenges to the experimental ones when describing these complex colloidal systems under dynamical conditions. For instance, recent experiments suggest that the boundary has a significant impact on the local structure of colloids[30–33] and on the equilibration pathways of structural relaxation of colloidal systems[34]; yet such understanding mainly comes from the examination of colloidal systems with static boundary conditions[30–35]. Moreover, one of the challenges of the current study is to disentangle the rate of self-assembly of the liquid crystalline tactoids from shape relaxation, providing insights on the kinetics of self-assembled complex colloidal systems.

Here we report the shape and structural relaxation dynamics of colloidal liquid crystalline tactoids. We use β-lactoglobulin amyloid fibrils and cellulose nanocrystals as model rod-like colloidal liquid crystalline systems. We disentangle shape and structural relaxation and show—by integrated experimental and numerical measurements— that the shape relaxation of the tactoids follows a single exponential decay that depends on the material isotropic and anisotropic properties and the size of the droplets. We develop a theoretical model to predict the shape of the tactoids out of the equilibrium state, by considering the Hamiltonian of the tactoids in presence of an external flow field. We also show that the structural relaxation of the tactoids follows different paths depending on the colloidal mesogens configurations at the ground state; homogenous and bipolar tactoids relax through a first-order exponential decay whereas cholesteric ones follow a second-order exponential decay. We use direct experimental measurements of the order parameter, supported by direct numerical simulation (DNS) of a complete structure-composition multiscale model, to discuss the nature of the structural relaxation of the liquid crystalline droplets and how it relates to the splay, bend and twist structures of tactoids at relaxed state. Our results offer original insights on the structural organization of colloidal suspensions out of equilibrium, under dynamic confined boundaries and evolving shapes.

## Results

**Relaxation of different classes of liquid crystalline droplets**. The classical approach to study relaxation of droplets involves using the four-roll mill geometry developed originally by Taylor[36]. This approach, however, is not applicable to tactoids due to their labile nature, making prohibitive isolating a single tactoid in such a geometry. Thus, in our experiments we take advantage of a microfluidic system with contraction-abrupt expansions design[37,38], allowing to elongate tactoids with different volumes and then let the elongated tactoids relax to the equilibrium state in the abrupt expansion zone (the details on the microfluidic channel are provided in Methods and Supplementary Note 1). To be able to form the tactoids in a microfluidic chip, we prepared the suspension of the liquid crystalline with a concentration that is set within the isotropic–nematic coexistence region. After injection of the suspension into the microfluidic system, tactoids with various volumes are formed inside the channel following nucleation and growth path. Series of experiments with β-lactoglobulin amyloid fibril and cellulose nanocrystals liquid crystalline droplets were performed and analyzed under crossed polarizers and LC (liquid crystal)-PolScope device, allowing capturing not only the shape of the tactoids but also their internal structure (Fig. 1), see Supplementary Movies 1–3. Regardless of the tactoids volume, all the tactoids at the initial state are extended and the director field inside the tactoids is aligned parallel to the long axis of the tactoid, known as homogenous configuration[2,4]. During the relaxation, an initially extended droplet with volume $V$ ($\approx r^2 R$ with $R$ the major and $r$ the minor axes of tactoids) $\sim 10^2$ μm³ undergoes the shape relaxation while its structure remains unchanged as homogenous configuration (Fig. 1a). As explained in detail later, while its configuration remains unchanged as homogenous, the tactoid still undergoes structural relaxation. A tactoid with a larger volume, $\sim 10^3$ μm³, shows relaxation on both its shape and structure where the director field changes from homogenous structure to bipolar with a director field that smoothly follows the tactoid interface (Fig. 1b). For a tactoid with volume $\sim 10^4$ μm³ as shown in Fig. 1c, while its shape relaxes to a nearly—yet not perfectly—spherical shape, the structural relaxation takes place while changing the director field from homogenous to cholesteric configuration, that is easily distinguishable from its characteristic striped texture.

We additionally captured both the shape and structural relaxation of the tactoids using DNS (see Supplementary Movies 4–6), and found good agreement with our experimental results as shown in Fig. 1. Details of the DNS are given in the Methods section.

**Shape relaxation of liquid crystalline droplets**. To characterize the underpinning physics of the relaxation of the liquid crystalline droplets, we first analyze the shape of the tactoids during the relaxation. We measured the long axis of tactoids at different time $t$ and quantified the relaxation behavior, as suggested previously for homogenous droplets[39], by $\mathcal{R} = \frac{R(t) - R_{\text{equil.}}}{R_{\text{init.}} - R_{\text{equil.}}}$, where $R_{\text{equil.}}$, $R_{\text{init.}}$, and $R(t)$ are the half-length of the long axis of tactoid at equilibrium, at the initial time, and at a given time, respectively (Fig. 2a–c). The value of $\mathcal{R}$ is one at time zero and zero when the tactoid reaches its equilibrium shape. We observed that $\mathcal{R}$ for all classes of tactoids (homogenous, bipolar, and cholesteric

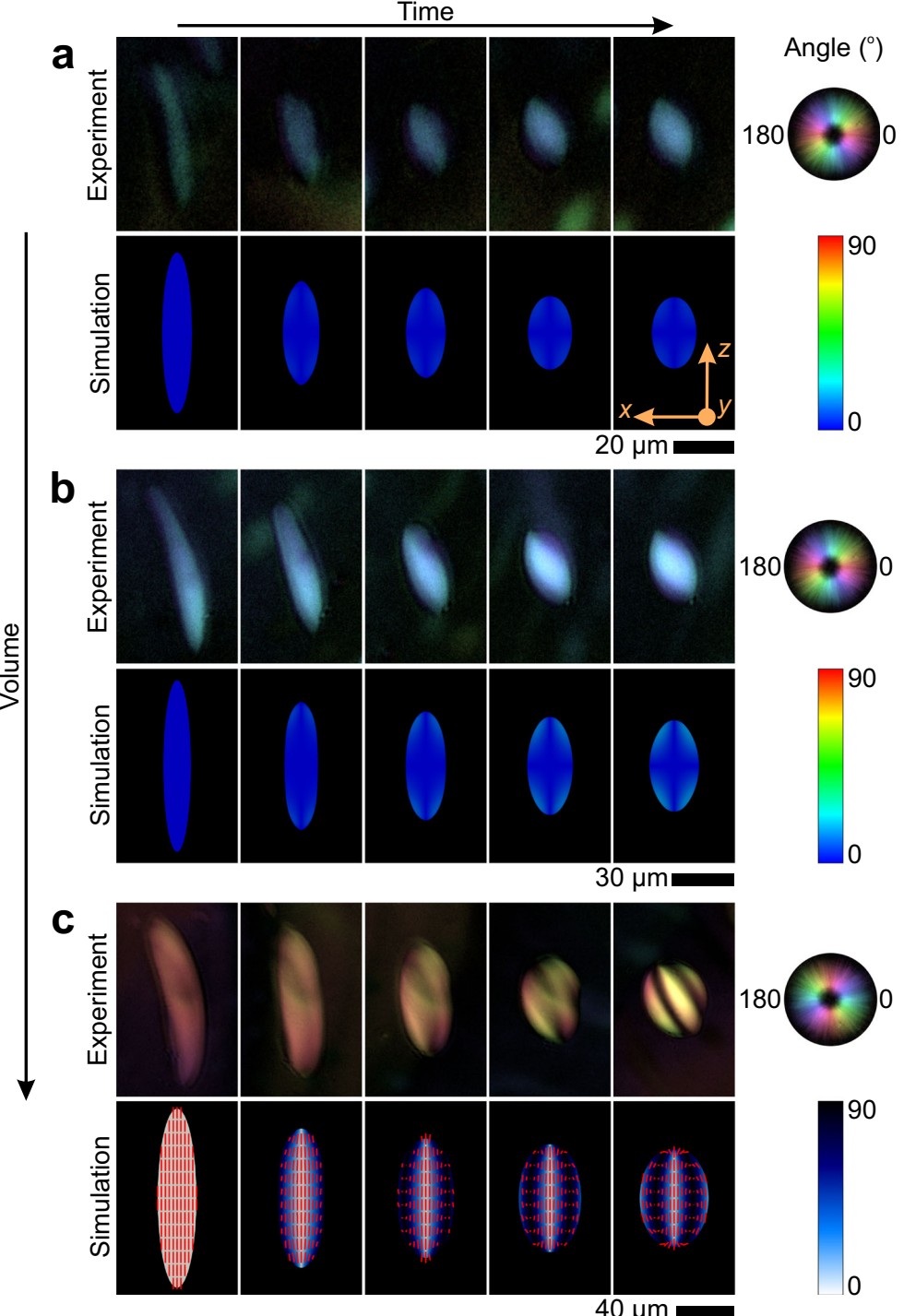

**Fig. 1 Relaxation of different classes of the amyloid fibril liquid crystalline droplets.** The sequence of time-lapse images of relaxation of initially extended amyloid fibrils liquid crystalline droplets with different volumes. In each panel, the first row shows the experimental results taken with LC (liquid crystal)-PolScope device. The colormaps corresponding to experimental results denote the orientation of the director filed in the x–z plane; the second row demonstrates the numerical simulation results with color bar capturing the director field orientation with respect to z-axis. The tactoids are at the homogenous configuration at the initial state and upon relaxation, they hold different configurations. **a** An initially extended tactoid with volume 644 μm³ undergoes shape relaxation while its configuration remains unchanged at homogenous configuration. **b** An elongated tactoid with volume 2751 μm³ relaxes both its shape and structure recovering a bipolar configuration upon relaxation. **c** A droplet with volume 16,414 μm³, having larger volume compared to (**a**, **b**), relaxes to a cholesteric structure with three bands. Since colors and director lines show the same information due to the axisymmetric nature of the homogenous (**a**) and bipolar (**b**) tactoids, the lines are not shown for better readability (see Supplementary Movies 4–6 for the version with lines). Note that the brightness of the experimental images is varied for better visualization.

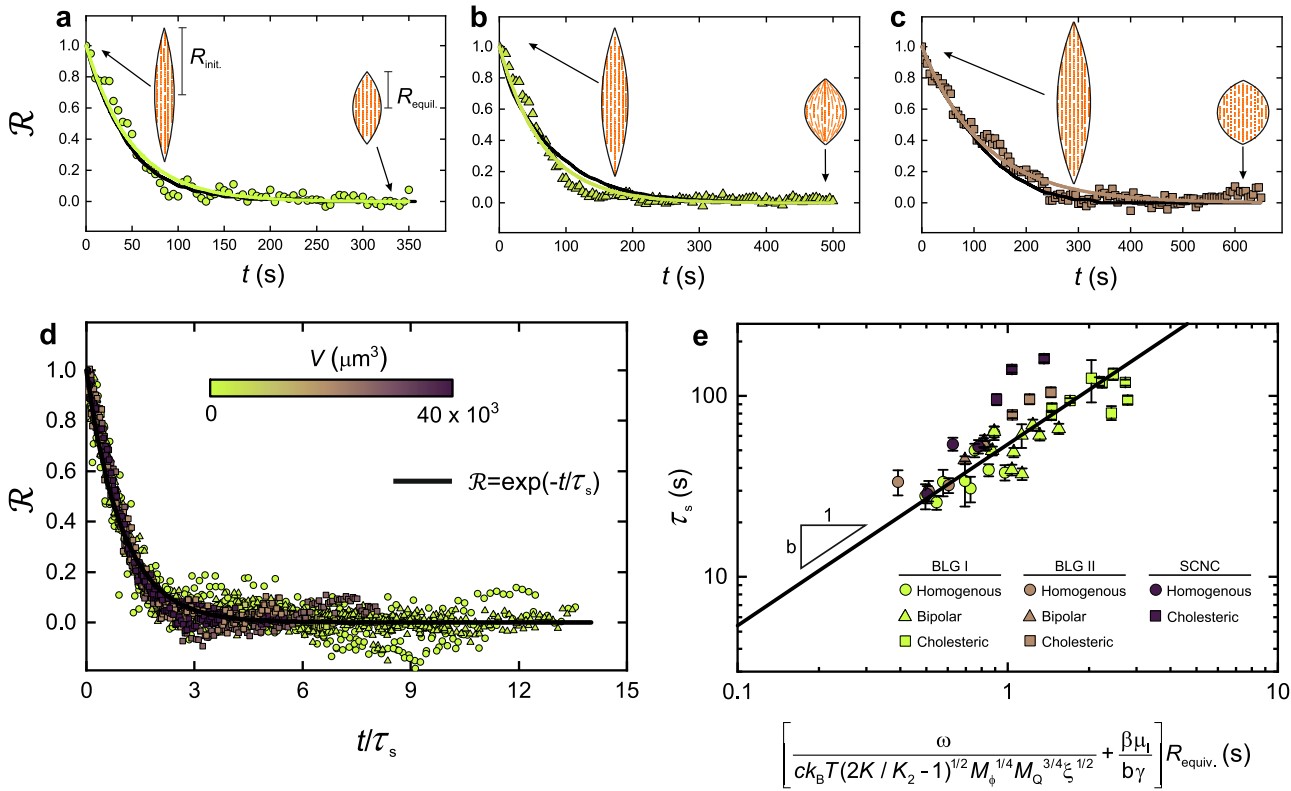

**Fig. 2 Shape relaxation of amyloid fibril liquid crystalline tactoids. a–c** Evaluation of $\mathcal{R}$ (defined as $\frac{R(t) - R_{equil.}}{R_{init.} - R_{equil.}}$ where $R_{equil.}$, $R_{init.}$, and $R(t)$ are the half-length of the long axis of tactoid at equilibrium, at the initial time, and at a given time $t$, respectively) with respect to time for tactoids that relax to homogenous, $R_{equiv.}$ ($V^{1/3}$, with $V$ the volume) = 9.4 μm (**a**), bipolar, $R_{equiv.}$ = 19.3 μm (**b**) and cholesteric, $R_{equiv.}$ = 27.8 μm (**c**) configurations at equilibrium. Symbols and black lines denote the experimental and numerical simulation results, respectively; colored lines show the fitting ($\mathcal{R} = \exp\left(-\frac{t}{\tau_s}\right)$) that is used to obtain the characteristic shape relaxation time, $\tau_s$. **d** Evaluation of $\mathcal{R}$ with respect to scaled time $t/\tau_s$ resulting in a universal curve, $\mathcal{R} = \exp\left(-\frac{t}{\tau_s}\right)$, for shape relaxation of the different classes of tactoids with various volumes and initial elongation values. **e** Circle, triangle, and square symbols denote homogenous, bipolar, and cholesteric tactoids, respectively. The error bars represent standard deviation. The developed theory, solid line, predicts the $\tau_s$ for different classes of BLG and SCNC liquid crystalline tactoids, confirming the generality of our approach to predict the bio-colloidal liquid crystalline tactoids relaxation behavior. Here, $\omega$ is the anchoring strength, $ck_B T$ is the thermal energy per unit volume of dispersion, $K$ and $K_2$ are the Frank elastic constants, $\xi$ is the coherence length, $M_\varnothing$ is the mass mobility, $M_Q$ is the rotational mobility, $\gamma$ is the interfacial tension, $\beta\mu_I$ is the effective viscosity, and b is a single constant pre-factor.

configurations) follows essentially a single exponential decay (Fig. 2a–c). Thus, to capture the shape relaxation behavior of the tactoids, we use $\mathcal{R} = \exp(-t/\tau_s)$, where the $\tau_s$ is the characteristic shape relaxation time of the tactoids. A good agreement between the fitting, using exponential decay $\mathcal{R} = \exp(-t/\tau_s)$, and the experimental data of the shape relaxation of the tactoids with different relaxed configurations, allows us to obtain the $\tau_s$ from the shape decay curve (Fig. 2a–c). Hence, a good collapse of the shape relaxation data of the tactoids with different volumes and relaxed configurations onto a single master curve is observed when time is rescaled with $\tau_s$ (Fig. 2d). We additionally performed experiments on the shape relaxation of the tactoids resulting from breakup events of initially extended tactoids in homogeneous configuration, and observed a single exponential decay trend similar to the initially extended tactoids relaxation behavior, see Supplementary Fig. 2.

To calculate $\tau_s$, we combine theories on simple droplet relaxation, dimensional analysis, and DNS that shows an excellent agreement with our experiments, see Figs. 1, 2a–c. As elaborated in Supplementary Note 2, we find:

$$\tau_s = b\left[\frac{\omega}{ck_B T(2K/K_2 - 1)^{1/2} M_\phi^{1/4} M_Q^{3/4} \xi^{1/2}} + \frac{\beta\mu_I}{b\gamma}\right]R_{equiv.} \quad (1)$$

where $\tau_s$ is expressed as the sum of two contributions: the first term is the liquid crystalline anisotropic contribution ($\tau_a$) induced

by the presence of colloidal mesogens accounting for orientational order, gradient elasticity, anisotropic viscoelasticity, rotational dissipation, and concentration gradients while the second term is the characteristic shape relaxation time of elongated isotropic tactoids ($\tau_i$). We express $\tau_i$, following the well-established relation of the characteristic shape relaxation time of elongated isotropic droplets[37,40,41], as $\tau_i = \frac{\beta\mu_I R_{equiv.}}{\gamma}$, where $\gamma$ is the interfacial tension, $R_{equiv.} = ((r^2 R)^{1/3})$ is the equivalent radius of tactoids, $\mu_I$ the viscosity of the medium phase -taken to be equal to viscosity of the isotropic phase- and $\beta = \frac{(2\hat{\eta} + 3)(19\hat{\eta} + 16)}{40(\hat{\eta} + 1)}$, where $\hat{\eta} = \frac{\mu_N}{\mu_I}$ is the ratio of viscosities of the nematic phase $\mu_N$ and that of the isotropic medium, $\mu_I$. We obtain $\tau_a$, as noted in Supplementary Note 2, based on dimensional analysis and parametric studies through our validated DNS on material properties involved in the liquid crystalline self-assembly and tactoid size. In Eq. 1, $\omega$ is the anchoring strength, which takes into account the coefficient of the concentration-orientation gradient and its coupling in the numerical simulations[42] (Supplementary Note 2), the term $ck_B T$ is thermal energy per unit volume of dispersion with $c$, $k_B$, and $T$, the number density, Boltzman constant, and temperature, respectively. The term $K$ is the Frank elastic constant for splay and bending (assumed to be equal) and $K_2$ is the Frank twist elastic constant. It should be noted that, from both theory[43,44] and experimental measurements[45–47] on

different systems of rigid rod-shaped liquid crystals, including filamentous colloids[48] analogous to those studies here, the ratio of $\frac{K}{K_2}$ in Eq. 1 is always greater than ½, and thus Eq. 1 only contains real arguments. The term $\xi$ represents the coherence length that is an indicator of length over which long-range ordering takes place. The term $M_\varnothing \propto \frac{\ln(L/D)}{c\mu_N L}$ is the mass mobility and $M_Q \propto \frac{2\ln(2L/D)-1}{c\mu_N L^3}$ is the rotational mobility[49] ($\propto$ stands for proportionality); here $L$ and $D$, respectively, the length and diameter of the rod-like mesogen, assumed to be equal to the weighted mean length of the fibrils $L_{f,w}$ and effective diameter $D_{eff.}$ proposed by Onsager[50], respectively. We list all properties of the liquid crystals used in this study in Table 1 along with the details on calculations and measurements in Supplementary Notes 3, 4, supported by refs. [51–59]. The relation for $\tau_a$ is formulated through a hybrid approach based on DNS results and dimensional analysis and can be turned to an equation by use of a single constant pre-factor ($b = 54.0$). The term $b$ is thus an aggregated value reflecting pre-factors present in the proportionality terms such as mobilities. We compare the experimental results for $\tau_s$ with our prediction with a single fitting parameter $b$ and find an excellent agreement (see Fig. 2e).

To test the generality of the present approach, in addition to BLG I in Table 1, we also perform experiments with one other system of amyloid fibrils (BLG II) having different length distribution compared to BLG I as a result different material properties[60] and with sulfated cellulose nanocrystals (SCNC); see Table 1. The tactoids of both systems also illustrate single exponential decay during the shape relaxation like BLG I in Fig. 2a–c. The results for characteristic shape relaxation time of BLG II and SCNC show very good agreement with Eq. 1 prediction (Fig. 2e), suggesting that Eq. 1 is general enough to describe the relaxation behavior of most bio-colloidal liquid crystalline tactoids.

**Deformation of tactoids under external stresses.** Having established a general picture on the dynamic shape relaxation of the tactoids, we go into modeling the deformation of the tactoids under external stresses. While this has been well documented for simple fluids[40,61], for liquid crystalline tactoids the physics become complex due to the energy terms associated with the internal structure of the tactoids, their anisotropic viscoelasticity and the confining boundary features. We look at the deformation of the droplet under uniaxial flow field with extension rate given $\dot{\varepsilon}_{xx} = \frac{\partial u_x}{\partial x}$, where the $u_x$ is the flow speed and $x$ is the direction of the motion of the flow, although the approach is general and can be used to model tactoids deformation under any external stresses. We consider the energy gained by the tactoids under the external stresses imposed by the extensional flow field and incorporate it to free-energy landscape of the tactoids that is well described by a scaling form of Frank–Oseen elasticity theory[2,4]. In particular, the rate of energy $\frac{dE}{dt}$ gained by the tactoids under any external normal stresses $\boldsymbol{\sigma}$ can be expressed as $\frac{dE}{dt} = \int \boldsymbol{\sigma} \cdot \mathbf{u}_i \, dS$, where $\mathbf{u}_i$ is the displacement of the interface of the tactoids. We consider the case where the tactoid is stretched due to the stresses applied by uniaxial extensional flow and in Supplementary Note 5

we calculate $\frac{dE}{dt}$ to be

$$\frac{dE}{dt} = -6\mu_I V\dot{\varepsilon}\frac{1}{r}\frac{dr}{dt}. \quad (2)$$

The free-energy landscape of the tactoids includes two energetic terms, the bulk elastic and surface free energies, associated with the tactoid at equilibrium. The total free energy of the tactoid $F_E$ is described in scaling form as[2,4]

$$F_E \sim \gamma Rr\left[1 + \omega(r/R)^2\right] + \frac{Kr^2}{R} + \frac{1}{2}K_2(\theta + q_\infty)^2 r^2 R, \quad (3)$$

where the first term accounts for the surface free energy of the tactoids that is due to the interfacial tension and the anchoring strength. The last two terms are the bulk elastic free energy of the tactoids where the second term accounts for splay and bending free energy and the third term is the twist elastic free energy. The term $\theta$ ($= n \cdot \nabla \times n$ with $n$ the nematic director) is the twist term in the Frank–Oseen elasticity theory and the term $q_\infty$ ($= 2\pi/P_\infty$ with $P_\infty$ the natural pitch of the system) is the chiral wave number. We measured and summarized the properties of the suspensions used in this study in Table 1 (see also Supplementary Note 4).

By energy conservation, the rate of the energy gained by the tactoids due to the normal stresses from the flow field must be equal to the rate of the energy changes in the free energy of the tactoids associated with their elastic/interfacial energy. Note that setting the two energies equal stands valid here since the process happens significantly faster than the rate at which the heat can flow out, thus separating the time scale for the transfer of the energy associated with structural changes from the time scale for heat dissipation. Additionally, as all three classes of the homogenous, bipolar, and cholesteric tactoids hold a homogenous configuration under extreme deformation, as can be seen in Fig. 1 and our recent study[9], we ignore the second term in Eq. 3, implying, as usual for homogeneous tactoids, that the bulk elastic energy due to the splay and bending is zero. Additionally, the third term is eliminated as, in the homogenous configuration and at constant tactoid volume, it does not change under deformation, so the rate of the energy gained by this term becomes zero. All in all, by setting Eq. 2 equal to the time derivative of Eq. 3 yields:

$$5\gamma\omega r^6 + 6\mu_I\dot{\varepsilon}V^2 r - \gamma V^2 = 0, \quad (4)$$

giving the steady-state elongated shape of the tactoids under a given extensional flow field in terms of $r$ as a function of $\dot{\varepsilon}$ and $V$. There is no analytical solution for Eq. 4, but the numerical solution along with our experimental data are presented in Fig. 3. Here, to best match experimental observations, the second term is rescaled by a pre-factor of 0.14, which is fully justified by the use of a scaling form of the Frank–Oseen energy landscape. Our results suggest that the short axis of the tactoids $r$ decreases as the extension rate increases, where $r$ lines, corresponding to tactoids with different volumes, converge to a single universal curve at large values of extension rate (Fig. 3a). The most remarkable consequence of our analysis is that for high extension rates, $r$ becomes independent of the volume, that is, the cross-section of the tactoid is simply ruled by extension rate, and that at identical extension rates, tactoids of different volumes $V$ only differ by

**Table 1 Amyloid fibrils and cellulous nanocrystals suspensions properties obtained with experiments or calculated.**

| Sample | $L_{f,m}$ nm | $L_{f,w}$ nm | $D_{f,m}$ nm | $D_{eff.}$ nm | $\mu_{I,0}$ Pa s | $\mu_{N,0}$ Pa s | $\gamma$ μN m⁻¹ | $\omega$ | $K_2$ pN | $K = K_3 = K_1$ pN | $P_\infty$ μm | $\xi$ μm |
|---|---|---|---|---|---|---|---|---|---|---|---|---|
| BLG I | 303 | 424 | 2.5 | 4.9 | 0.12 | 0.06 | 1.1 | 1.17 | 1.6 | 8.2 | 25.6 | 1.3 |
| BLG II | 245 | 371 | 4.2 | 5.9 | 0.12 | 0.06 | 0.8 | 1.00 | 1.3 | 5.7 | 30.3 | 1.0 |
| SCNC | 162 | 200 | 4.5 | 15.7 | 0.20 | 0.20 | 1.3 | 0.65 | 0.4 | 1.2 | 9.0 | 0.9 |

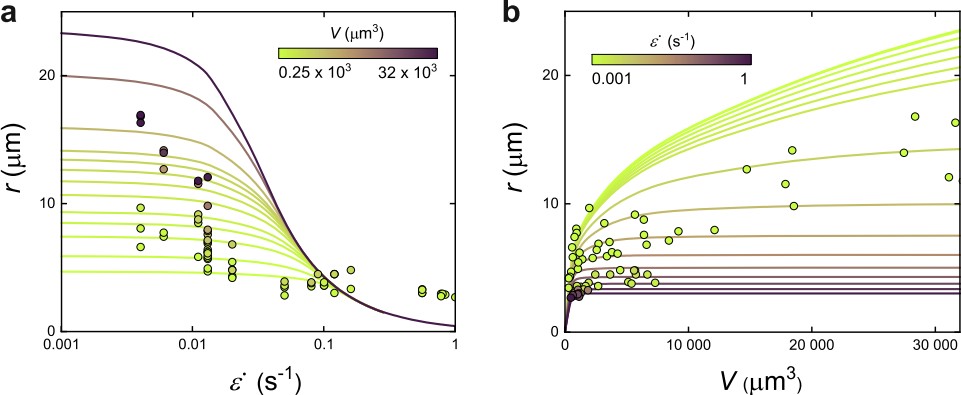

**Fig. 3 Maximum deformation of tactoids under various extension rate. a** The theory (lines) and the experimental data (symbols) predict that short axis of the tactoids $r$ decreases as the extension rate $\dot{\varepsilon}$ increases where $r$ lines, corresponding to tactoids with different volumes $V$, converge to a single curve at large values of extension rate. **b** At a given extension rate, the short axis of the tactoids increases logarithmically with an increase in the volume of the tactoids.

their long radius $R$, which is directly proportional to $V$. In the regime of low extension rate, the short axis of the tactoids becomes volume-dependent and increases logarithmically with the increase in the volume of the tactoids (Fig. 3b), which is nicely supported by the collected experimental dataset. Equation 4 can be used to predict the $R_{init.}(=V/r^2)$ as the maximum tactoids deformation that can be reached under a given extensional flow rate. Note that in Fig. 3a, b the experiments of the large volume of the tactoids (e.g., $V = 30,000\,\mu m^3$) at the high extension rate is limited by the experimental setup. This is so as the length of the tactoids in the stretched forms becomes extremely high compared to the size of taken images, preventing to fully capture tactoids with large volume at high extension rate.

**Structural relaxation of nematic-cholesteric tactoids.** The above treatment describes comprehensively the evolution of the confining boundaries of the tactoids, but provides no information on the evolution of their internal structure. We thus turn our attention to the study of the orientational order parameter of the director field during relaxation of the tactoids. In our experiments, we captured the relaxation of the tactoids by LC-PolScope allowing us to access the retardance images of the tactoids during relaxation and analyze the order parameter $S$, where $S = \imath/d\Delta n_0$ with $\imath$ the optical retardance value, $d$ the thickness of the sample and $\Delta n_0$ the birefringence corresponding to a perfectly aligned nematic phase, i.e., when the order parameter is 1[62,63]. We measured the retardance value $\imath$ of every pixel within the tactoids, and accordingly, the $d$ is calculated for every pixel assuming a spindle shape for the tactoids (see Supplementary Note 6). The exact value of the $\Delta n_0$ is unique for a given liquid crystalline system and is often challenging to obtain experimentally, thus here we present our calculation independent form $\Delta n_0$. We define $S = \frac{S(t)-S_{equil.}}{S_{init.}-S_{equil.}}$ to capture the structural relaxation of the tactoids, similar to the one used for the shape relaxation of the tactoid, and most importantly, fully independently of $\Delta n_0$. The experimental results of $S$ obtained for tactoids with different relaxed configurations against scaled time are shown in Fig. 4a–c. The structural relaxation of the tactoids with small volumes that hold homogenous and bipolar at equilibrium follows a first-order exponential decay. However, for the larger extended droplets that relax to cholesteric structure, $S$ shows non-monotonic behavior where initially $S$ decreases until its minimum, then it starts to increase before reaching its equilibrium, indicating a second-order exponential decay (Fig. 4c). To illustrate such a undershoot behavior more clearly, while maintaining $S$ as a

positively-defined object, we use in this case $S = \frac{S(t)-S_{minimum}}{S_{init.}-S_{minimum}}$ and show the rate of the changes in $S$ versus the scaled time in Fig. 4d–f (note that the two definitions of $S = \frac{S(t)-S_{minimum}}{S_{init.}-S_{minimum}}$ and $S = \frac{S(t)-S_{equil.}}{S_{init.}-S_{equil.}}$ are equivalent for homogeneous and bipolar tactoids following a monotonic decay). It is clear from Fig. 4 that numerical simulations capture the first-order exponential decay in the case of homogenous and bipolar tactoids. In the case of cholesteric tactoids, our numerical simulation results bear qualitatively similar behavior for the structural relaxation of the tactoids although simulations underestimate the equilibrium order parameter obtained experimentally for cholesteric tactoids structural relaxation (Fig. 4c). To illustrate this further, we show the rate of the changes in $S$ versus the scaled time in Fig. 4d–f. Quantitatively, the tactoids with the homogenous and bipolar configuration at relaxed state follow $S = \exp(-t/\tau_c)$, where we experimentally find the characteristic structural or configurational relaxation time $\tau_c$ to be 24.4 s and 31.9 s for homogenous ($V = 644\,\mu m^3$) and bipolar ($V = 2751\,\mu m^3$) tactoids, respectively. In contrast, the tactoid with cholesteric configuration at equilibrium ($V = 16,414\,\mu m^3$), shows two characteristic configurational relaxation times captured by $S = c_1\exp\left(-t/\tau_{c,1}\right) + (1 - c_1)\exp(-t/\tau_{c,2})$, with $c_1$ a constant equal to $-1.6$, and $\tau_{c,1}$ and $\tau_{c,2}$ found to be 105.9 s and 333.7 s, respectively.

To interpret the physics behind these structural relaxations, we inspect the time scale related to bending, splay and twist terms in tactoids. Compared to the relaxation to homogeneous/bipolar tactoids, which involves nematic ordering with at best splay/bend relaxation, when the configuration relaxes to cholesteric tactoids, an additional twist relaxation takes place (see numerical simulation results in Supplementary Movies 4–6). This suggests that the second exponential decay associated with the cholesteric tactoids originates from the twist term. We suggest that in $S = c_1\exp\left(-t/\tau_{c,1}\right) + (1 - c_1)\exp(-t/\tau_{c,2})$, where $\tau_{c,2}$ is significantly longer than $\tau_{c,1}$, the second exponential decay $\tau_{c,2}$ originates from the chiral twist term while $\tau_{c,1}$ originates from simple nematic ordering. We base this statement on two grounds. First, we compare the length-scales of the nematic and cholesteric ordering. We consider the length scale of the cholesteric phase as the length which is required for the phase to form a single periodic twist, that is set by the inverse wave number (i.e., the pitch) which is in the order of $10^1\,\mu m$. In contrast, to form nematic ordering (characterized by splay and bending), the length scale is defined in the range of the length of the fibrils (mesogens)

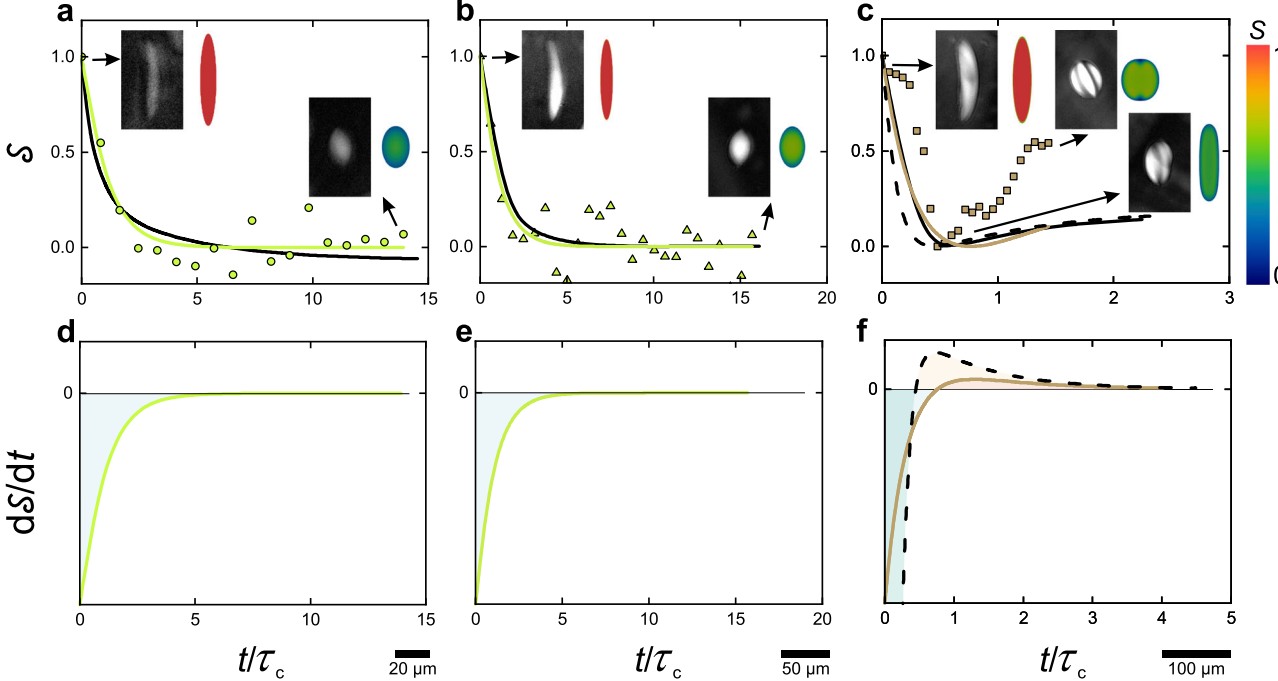

**Fig. 4 Structural relaxation of different classes of liquid crystalline tactoids. a–c** Evaluation of $\mathcal{S}$ (defined as $\frac{S(t)-S_{\text{equil.}}}{S_{\text{init.}}-S_{\text{equil.}}}$ for homogeneous and bipolar tactoids and $\mathcal{S} = \frac{S(t)-S_{\text{minimum}}}{S_{\text{init.}}-S_{\text{minimum}}}$ for cholesteric tactoids, where $S_{\text{equil.}}$, $S_{\text{init.}}$, and $S(t)$ are order parameter values at equilibrium, at the initial time, and at a given time $t$, respectively) with respect to scaled time, $\frac{t}{\tau_c}$ with $\tau_c$ the characteristic configurational relaxation time, for tactoids that relax to homogenous (**a**), bipolar (**b**) and cholesteric (**c**) configurations at equilibrium. The experimental insets showing the retardance images taken with LC-PolScope along with numerical simulation results present the critical state of the relaxation for each class of the tactoids. Color bar denotes the order parameter values in numerical simulation insets. Note that the brightness of the experimental images is increased for better visualization. The symbols denote the experimental data, black solid lines are numerical simulation results. Colored and dashed black lines show the fitting that is used to obtain $\tau_c$ from experimental and numerical simulation results, respectively. **d–f** The changes in $\frac{d\mathcal{S}}{dt}$, obtained from the fitted lines in **a–c**, during relaxation for different classes of tactoids: homogenous (**d**), bipolar (**e**), and cholesteric (**f**) configurations. While homogenous and bipolar tactoids follow monotonic single exponential decay during relaxation $\mathcal{S} = \exp\left(-\frac{t}{\tau_c}\right)$, the cholesteric tactoids are characterized by a non-monotonic behavior of $\mathcal{S}$ during relaxation (see (**c**)), described by a second-order exponential decay, $\mathcal{S} = c_1\exp\left(-t/\tau_{c,1}\right) + (1 - c_1)\exp(-t/\tau_{c,2})$, where $c_1$ is a constant.

which is in the order of $10^{-1}$ μm. Thus, we argue that the larger twist length-scales compared to splay/bend imply longer relaxation times for the twisting deformation. Secondly, from the experiments of the relaxation of the cholesteric tactoids using a LC-Polscope, allowing us to capture the changes in the director field over time and follow the twist dynamics in the director field (see Supplementary Fig. 6), we find that change/rotation in the director field (the twist) takes place until the latest stages of the relaxation process and the twist changes are significant in the latest stage, suggesting again that the second exponential decay $\tau_{c,2}$ originates from the twist re-arrangement. It is also worth mentioning that the structural relaxation time in tactoids is much lower than under static fixed boundary conditions. The structural relaxation time is of the order of hours for similar BLG and SCNC systems inside a capillary tube[35] as opposed to tens/hundreds of seconds here. It is indeed known that boundary mobility increases the mobility of the mesogens thus promoting faster kinetics in structural relaxation[12].

What is the configuration of the extended tactoid with a given initial volume after relaxation? We are able to predict the relaxed configuration of the tactoids using the theoretical modeling recently developed starting from a scaling form of Frank–Oseen elasticity theory[4]. According to this theory, the tactoids at equilibrium hold homogeneous configuration when $(V/\alpha)_{\text{Homogenous}} < (K/\gamma\omega)^3$, bipolar configuration when $(K/\gamma\omega)^3 < (V/\alpha)_{\text{Bipolar}} < [1.7\gamma/(K_2q_\infty^2)]^3$, and cholesteric configuration when $(V/\alpha)_{\text{Cholesteric}} > [1.7\gamma/(K_2q_\infty^2)]^3$. Approximating α equal to 3 for homogenous-bipolar and 1.5 for bipolar-cholesteric boundaries following ref. [4], we computed these

threshold values for BLG I and found $V_{\text{Homogenous}} \lesssim 800$, $800 \lesssim V_{\text{Bipolar}} \lesssim 11,000$, and $V_{\text{Cholesteric}} \gtrsim 11,000$ μm³. This confirms that the tactoids shown in Fig. 1 follow a relaxation path until equilibrium. Thus, knowing the initial volume of the tactoids, their configuration after relaxation can be predicted simply from the scaling form of Frank–Oseen elasticity theory and physical parameters of the system such as elastic constants, interfacial energy and anchoring strength[4]. We provide the nematic-cholesteric phase diagram of the tactoids collected from the samples in a cuvette at equilibrium showing tactoids configuration as a function of the volume in Supplementary Note 7, as a further demonstration that the initially stretched tactoids reach an equilibrium configuration after relaxation.

We have presented an integrated picture based on experiments, numerical simulations and theory, allowing a disentanglement and comprehensive description of the shape and structural relaxation of initially stretched colloidal liquid crystalline droplets. We have shown that these tactoids undergo relaxation on both shape and structure, when the external flow field maintaining them in a non-equilibrium state is released. Independently of the size, the shape relaxation of the tactoids is characterized by a single exponential decay, which is explained well by taking into account both isotropic and anisotropic features of the tactoids. In contrast, the structural relaxation follows different fates, with first- and second-order exponential decays, depending on the existence of splay, bend and twist contributions in the ground state, whose relative weight depend directly on the size of the tactoid. We have discussed

the fundamental physical mechanisms behind the shape and structural relaxation and highlighted their interdependence. These results bring forward our understanding of dynamic processes in liquid crystalline systems based on filamentous colloids and introduce a combined experimental, theoretical and numerical formalism which can be extended to heterogeneous complex fluids, soft matter and biological colloids in general.

## Methods

**Amyloid fibrils liquid crystals**. β-lactoglobulin was purified from whey protein following ref. [64] and dissolved in Milli-Q water at 2 wt%. The solution was cleared from aggregates by filtering using 0.45 μm Nylon syringe filter and pH of the solution was adjusted 2 by adding HCl. Later, the solution was heated for 5 hrs over a hot plate at 90 °C. Once the amyloid fibrils were prepared, we shortened the length of fibrils using the mechanical shear force method. Two sets of the solutions were prepared, see Supplementary Note 3 for details on length and height distributions. The solutions were dialyzed for 5 days using 100 kDa MWCO Spectra/Por dialysis membrane against pH 2 Milli-Q. The bath was changed every 24 h. To reach isotropic–nematic coexistence region concentration for the solution, suspensions up-concentrated using 6–8 kDa MWCO Spectra/Por 1 dialysis membrane against 6 wt% polyethylene glycol solution (mol wt: $M_r$ ~ 35,000, Sigma Aldrich) in pH 2 milli-Q water. The solutions were kept in the fridge until phase separation happens, allowing reporting the isotropic and nematic phases concentrations in Table 1.

**Cellulose nanocrystal solution**. Cellulose nanocrystal suspensions were prepared by mixing freeze-dried cellulose nanocrystal (FPInnovations) in Milli-Q water. To make sure that cellulose nanocrystal dispersed well, the solution was ultrasonicated for 120 s. This was followed by centrifugation for 20 min at $12,000 \times g$ to remove aggregates. SCNC solution with concentration within the isotropic–nematic coexistence region was obtained by initially mixing 2.5 wt% freeze-dried cellulose nanocrystal in Milli-Q water.

**AFM measurement**. To perform the AFM measurement, a droplet of diluted suspension (0.01 wt%) was deposited on freshly cleaved mica. After 2 min, the mica was rinsed with Milli-Q water and dried with air stream. The images of the sample were captured at ambient conditions with MultiMode VIII scanning probe microscope (Bruker) in tapping mode. The software FiberApp[65] was used to analyze the images and measure the length and height distributions of fibrils.

**Microscopy measurement**. We performed the experiments using an optical microscope Zeiss equipped with crossed polarizers and combined with LC-PolScope universal compensator. Under crossed polarizers, time-series images at the frame rate of 12 frames per minute were taken. The microfluidic channel was placed on the microscope in a way that the tactoids long axis held 45° angle with respect to one of the crossed polarizers. This allowed unambiguous measurement of the tactoids short and long axis during relaxation. To perform the measurements, MATLAB program and ImageJ software were used. Furthermore, we used optical microscopy combined with LC-PolScope universal compensator at time-series mode capturing 3 frames per minute. LC-PolScope images were used to analyze the internal structure of the tactoids. Additionally, LC-PolScope produces the retardance images giving the retardance value of the image, pixel by pixel, which were used to measure the order parameter.

**Microfluidic**. The classical soft lithography approach was employed to make microfluidic systems[66]. We made PDMS by mixing polydimethylsiloxane (PDMS) monomer and curing agent (Dow Corning Slygard 184) with ratio 10 to 1. The plain glass slide (Corning 2947) was used as the base plate to attach the PDMS channel.

We used microfluidic system with rectangular cross-section with width of the channel at expansion zone $w_e = 600$ μm, the width of the contraction zone $w_c = 50$ μm, and the height of the channel $h = 100$ μm (see Supplementary Information for the schematic of the microfluidic system).

**Experimental details**. We performed all the experiments at room condition. The equipment used to run experiments in microfluidics consists of a Harvard Apparatus syringe pump, 250 μl Hamilton syringe, flexible tubing with inner diameter 0.8 mm, and the needle with inner diameter 0.34 mm and outer diameter 0.64 mm.

**Direct numerical simulations**. Past studies[23,28,67–69] have shown that the time-dependent Ginzburg–Landau model can capture the spatio-temporal coupled relaxation dynamics of shape and structure. Furthermore, our approach can result in high fidelity simulations capturing dynamics of spatio-temporal liquid crystalline self-assembly including breakage, coalescence, and defect evolution. However, these are out of scope of the current study. In the present study, we applied this

modeling approach and we focused on the relaxation dynamics of the shape and structure in initially extended tactoids, which are isolated in isotropic phase. The implementation has been elaborated in detail in our previous works, see refs. [23,28,67]. At the initial time, we consider an elongated tactoid according to the experimental observations. Thereafter, we let the elongated tactoids relax. Through relaxation, the total free energy is minimized according to the time-dependent Ginzburg–Landau model, by which the excess free energy such as surface tension and elasticity are relieved. The elongated tactoid self-selects the equilibrium shape and structure through a spontaneous thermodynamic-driven relaxation. Furthermore, the matrix surrounded the tactoid under relaxation is essentially kept at the isotropic concentration. Note that, in present work, we study three prime liquid crystalline configurations; homogeneous nematic, bipolar nematic, uniaxial cholesteric, which are all fully rotationally symmetric[70]. Given this fact, to reduce computational costs, we rely on rectangular two-dimension simulations which provide a good 3D description since there is no need to discriminate between point, line and ring disclinations; see Supplementary Movies 4–6.

## Data availability

The data that support the findings of this study are available from the corresponding author upon request

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

## Acknowledgements

We thank X.Cao (ETHZ) for help in microfluidic chips fabrications and Samuel Mathews for support to maintain in-house high-performance supercomputer. We thank Prof. Andrew de Mello (ETHZ) for granting access to his laboratory and Y.Yuan for helpful discussions. This work is supported by Sinergia grant no. CRSII5_189917 from the Swiss National Science foundation (R.M.).

## Author contributions

H.A. and R.M. conceived and initiated the project, designed the experiments, analyzed the data, developed theoretical modeling of the tactoids deformation, contributed to the theoretical analysis of relaxation time of the tactoids, and wrote the bulk of the paper. H.A. built the experimental apparatus and performed the experiments. M.B. contributed to the experiments and carried out the AFM measurements. S.A.K. and A.D.R. designed the simulations, analyzed the data, developed the theoretical analysis of relaxation time of the tactoids, and contributed to the writing of the paper. S.A.K. performed the simulations. A.D.R. and R.M. supervised the research. All authors discussed and edited the paper.

## Competing interests

The authors declare no competing interests.
