## [Peer review file · Nature Communications]

REVIEWER COMMENTS

Reviewer #1 (Remarks to the Author):

This article reports on the shape and structural relaxation of colloidal tactoids. The work involves a rather impressive combination of experiment, theory and computer simulation. The result yields a comprehensive understanding of the relaxation process. This represents a significant advance that justifies publication in nature communications.

Tactoids are a rather new and novel liquid crystal based colloid. While their applications are still limited, I believe they will form the basis of novel technology in the future. Similar to the extensive studies in the 1990s of vesicles formed from lipid membranes which form the basis of the understanding of encapsulation of mRNA vaccines today, I believe studies like this one will form the basis for future technology.

There are a couple of points that I think could improve the paper:

1. The introduction could be better written to appeal to a broader audience. The topic naturally mixes terms from different fields that a broader audience may not be familiar with. Simplifying the language in this section could be helpful.
2. The the paragraph after Eq.(3) the authors appeal to energy conservation. However, this is a viscous fluid in contact with its surroundings so energy conservation is not strictly followed here. I think the process being described here, though, happens fairly quickly compared to the rate at which heat can flow out so it is the separation of time scales for these processes that makes the authors argument valid here. A rephrasing, or footnote, could be useful here to clarify the argument.
3. I believe you are allowed 70 references in the main article where you have only 50. You should move 20 of the references in the supplementary material to the main references. (This ensures they are picked up by citation indexes which makes your article easier to find in a "forward" reference search so benefits you as well as the authors of these articles).

Reviewer #2 (Remarks to the Author):

By experimental work combined with numerical modeling and simulations, the authors investigate the relaxation of tactoids that were brought out of equilibrium by subjecting them to an elongational flow.

The initially elongated nematic liquid crystalline droplets relax both in terms of shape (eventually assuming the characteristic lens-shaped form) as well as, under particular conditions, in terms of internal structure, where, from homogenous alignment, bipolar or cholesteric configurations can emerge. The manuscript reports that the shape relaxation follows an exponential decay (the experimental results being supported by the numerical results). For the structural relaxation the authors find that a first order decay properly describes two of the cases (i.e. where the equilibrium configuration is either homogenous or bipolar), whereas for the cholesteric equilibrium tactoids a second order exponential decay is necessary. The numerical simulation nicely agrees with the experimental finding only for the first two cases.

The submitted manuscript is a follow-up of the recent paper by part of the authors of the current work (ref 9, published as well in Nature Communications). In both works (i.e. this manuscript and ref 9), the behavior of amyloid fibril tactoids under flow and their relaxation is investigated using a microfluidic setup. In ref 9, the focus is set on the shape changes and internal order transitions of the tactoids, whereas here the emphasis lays more on the relaxation mechanism. The current work features as well a numerical modeling of the relaxation. Another appealing addition compared to the previously published work is the two experimental systems (a different preparation of amyloid fibrils and a cellulose based liquid crystalline material) which gives more generality to the work.

From the perspective of the topics tackled in this manuscript, this work is more specialized than the previous paper (ref 9) and therefore less susceptible for a larger audience. Even if the paper is generally well written, the introduction does not give a convincing picture of why the topic investigated here is a crucial step forward in liquid crystal physics or what the implications or uses of the findings in this manuscript are. A more comprehensive range of applications would have probably strengthened the claim of the authors.

One aspect that needs to be clarified concerns the breakage and merging of tactoids. These two types of events are briefly mentioned in the previous publication (ref 9), but are completely absent from this manuscript. I would expect that tactoids that have resulted from, for example, a coalescence event would have quite a different equilibration dynamic both in terms of shape as well as structure and therefore I guess the authors choose not to present the corresponding data in this manuscript. What is the probability of occurrence of a breakage of a tactoid in two or more smaller size components or two tactoids merging?

Furthermore, coalescing of nematic liquid crystalline droplets often results in tactoids containing topological defects that arise from the mismatch between the directions of alignment of the merging tactoids. Two examples out of the literature dealing with this problem are the following:

Kim, Young-Ki, Sergij V. Shiyanovskii, and Oleg D. Lavrentovich. "Morphogenesis of defects and tactoids during isotropic–nematic phase transition in self-assembled lyotropic chromonic liquid crystals." *Journal of Physics: Condensed Matter* 25, no. 40 (2013): 404202.

Zhang, Chiqun, Amit Acharya, Noel J. Walkington, and Oleg D. Lavrentovich. "Computational modelling of tactoid dynamics in chromonic liquid crystals." *Liquid Crystals* 45, no. 7 (2018): 1084-1100.

Have the authors observed such a behavior in their experiments? If so, could the authors comment on the equilibration of such defects-containing tactoids?

The modeling presented in the manuscript cannot capture, at least in the present form, events like breaking or merging of tactoids. Would it be possible to extend it to capture such events? Could the authors comment on this aspect?

The relaxation of tactoids has been already investigated, however in a different set-up, where confinement is pre-imposed in the following paper:

Gârlea, Ioana C., Oliver Dammone, José Alvarado, Valerie Notenboom, Yunfei Jia, Gijsje H. Koenderink, Dirk GAL Aarts, M. Paul Lettinga, and Bela M. Mulder. "Colloidal liquid crystals confined to synthetic tactoids." *Scientific reports* 9, no. 1 (2019): 1-11.

There, the authors report a variety of different equilibration pathways which result from a mismatch between initial orientation of the liquid crystal and confinement-imposed orientation. Could the authors comment on the results in this paper in connection with their findings?

Considering the above remarks, I would not recommend publications in *Nature Communication* (at least not in the current form).

Minor remark:

Figures are sometimes hard to read due to crowding of elements, hindering the reader to properly see the results. One example is Figure 1, where, in the Simulation panels, in addition to color-coded director field orientation (in a particular plane), the orientation given by small red lines is shown. Due to the very dense mesh, it is: firstly, impossible to properly distinguish the orientation of these lines (at least in a printed size of the figure) since they are much too small and, secondly, these lines partially hide the director field below. I suggest decreasing the density of lines which would allow for size increase or removing them completely (for panel a and b they do not provide additional information).

Another example is Figure 2, panel e, where a number of very similar markers both in shape and color are used. A different choice of symbols/colors would help.

Reviewer #3 (Remarks to the Author):

The authors study the relaxation of the shape and liquid crystal structure of externally elongated tactoids towards equilibrium (both experimentally and numerically). They show that for different volumes of the tactoids, the shape relaxation is followed by an exponential decay towards the equilibrium. The time scale of this decay has isotropic (similar to simple droplets) and anisotropic (due to the liquid crystalline structure) contributions.

The relaxation of the nematic order parameter, however, has first and second-order exponential decays, depending on the ground state of the liquid crystal.

In addition to experiments, the following theoretical approaches are used to understand the experiments:

- 1-The time scale of the shape relaxation is found using dimensional analysis.
- 2- Using energy conservation, the authors find the initial shape of the tactoid due to the external flow.
- 3- Finally, numerical simulations have been used to study the evolution of the liquid crystal.

The question that has been tried to answer is interesting, however, there are some points that are very unclear to me (see the comments below). If the authors can precisely address all these points, I would be happy to recommend the manuscript for publication.

1- In eq. 1 for the time scale of the shape relaxation, if bend/splay elastic coefficient becomes larger than the twist elastic coefficient ($2K < K_2$) the time scale becomes imaginary. Can the authors comment on this? Also, due to the presence of the term $(2K/K_2 - 1)^{0.5}$ in the denominator, the timescale can diverge when $2K/K_2$ goes to 1. Could the authors please clarify what this means?

2- I suggest that the authors compare the experimental data on the initial shape of the tactoid with the solution of eq. 4 (adding data points to Fig. 3). Without a comparison, equation 4 is a mathematical work-out that has not been used throughout the paper.

3- Page 9, last paragraph:

It is clear from the experimental data there are two time-scales. But more explanation is needed here. Does the above paragraph mean that the system first relaxes to a nematic phase and then the cholesteric phase starts to appear? If so, why does the system first go to a nematic phase instead of directly going to a cholesteric phase?

Also, what do they mean by “bend/splay and twist occur at different length-scale”? Why are these length scales different?

4- In Fig. 4, it would be helpful to show the two exponential fits for S in Part C of the figure.

5- The simulations of the evolution of the LC seem to fail to capture the decay towards the cholesteric phase (Fig. 4C). This has not been referred to throughout the text. If the model is generic enough, it should be able to capture the dynamics towards the cholesteric phase too.

6- In experiments, is it possible to look at the evolution of the bend, splay, and twist distortions towards equilibrium? A graph with measurements in the elastic free energy of these deformations over time could be helpful in understanding the dynamics towards equilibrium.

Minor comments:

a) Page 8, second line: both the second and the third terms have been ignored from Eq.

b) What is the final shape of the tactoid? Does it relax to a sphere or a spindle?

c) It is not clear how the set-up in Fig. S3 is related to the cylindrical coordinate used in Eq. S16. I suggest that the authors use a 3D schematic figure to show the 3d experimental set-up. In addition, the region in which the drop elongates under the flow and the region where it relaxes is not clear. The expansion and compression regions are marked but it is not clear how the set-up works. A would suggest using a schematic time series of the different stages of the experiment.

d) In Eq. S23 why are the bend splay free energy proportional to area (r^2) and twist is proportional to volume (r^3)?

Reviewer #1

We would like to thank the Reviewer #1 for his/her comments on our manuscript. Below is our response with references to the pages of the revised manuscript.

This article reports on the shape and structural relaxation of colloidal tactoids. The work involves a rather impressive combination of experiment, theory and computer simulation. The result yields a comprehensive understanding of the relaxation process. This represents a significant advance that justifies publication in nature communications.

Tactoids are a rather new and novel liquid crystal based colloid. While their applications are still limited, I believe they will form the basis of novel technology in the future. Similar to the extensive studies in the 1990s of vesicles formed from lipid membranes which form the basis of the understanding of encapsulation of mRNA vaccines today, I believe studies like this one will form the basis for future technology.

There are a couple of points that I think could improve the paper:

1. The introduction could be better written to appeal to a broader audience. The topic naturally mixes terms from different fields that a broader audience may not be familiar with. Simplifying the language in this section could be helpful.

As the Reviewer pointed, in the revised manuscript, we modified the Introduction to simplify the language to make it available for a broader audience.

We modified on page 2:

“Colloidal liquid crystals are a class of soft matter formed when shape-anisotropic nanoparticles are dispersed in an isotropic fluid¹. When confined to a finite volume, rod-like colloidal particles self-organize into various structures that are set by a delicate balance between anisotropic viscoelastic and surface properties²⁻⁴. The subtle balance between these contributions results in facile response to external fields^{1,5} such as mechanical, flow, electric, and magnetic, giving rise to many opportunities and technological applications. Example includes displays, spatial light modulator and tunable filters in medical devices and optics, liquid crystal biosensors for rapid diagnostics, and new functional material such as artificial muscles exploiting liquid crystalline anisotropic physical properties⁶⁻⁸. However, facile responsiveness to external fields (and disturbance) makes them very fragile to be studied experimentally under dynamical conditions^{5,9}. In particular, relaxation of liquid crystalline droplets under mobile

confinement is still poorly understood despite its central importance in a variety of phenomena in condensed matter physics. This includes particle packing¹⁰, self-assembly¹¹, and relaxation of colloidal liquids¹² with implications in the field of active nematic, e.g. living liquid crystals¹³, where the understanding of the hydrodynamics of the liquid crystals is critical¹⁴.

Liquid crystalline droplets, known as tactoids, are a particularly significant example of colloidal liquid crystals, since they consist of micro-confined liquid crystalline colloids with a self-selected shape/structure stemming out from the thermodynamically-driven phenomena from which they emerge, i.e. spontaneous liquid-liquid crystalline phase separation (LLCPS)¹⁵⁻²³. In stark contrast with spherical liquid crystalline emulsions, achieved commonly by emulsifying liquid crystals in another immiscible liquid (like water in oil)²⁴⁻²⁵, tactoids hold spindle-like, prolate, or oblate shapes with different nematic-cholesteric internal structures^{2-4,15-22,26}, as a consequence of the coupling between the vanishingly small interfacial tension, the surface anchoring at the interface, the chirality of colloids and the anisotropic elastic properties^{3,4}. These features make tactoids a very unique system with peculiar viscoelastic²⁷⁻²⁹ and boundary^{3,4} properties, thus adding theoretical challenges to the experimental ones when describing these complex colloidal systems under dynamical conditions. For instance, recent experiments suggest that the boundary has a significant impact on the local structure of colloids³⁰⁻³³ and on the equilibration pathways of structural relaxation of colloidal systems³⁴; yet such understanding mainly comes from the examination of colloidal systems with static boundary conditions³⁰⁻³⁵. Moreover, one of the challenges of the current study is to disentangle the rate of self-assembly of the liquid crystalline tactoids from shape relaxation, providing insights on the kinetics of self-assembled complex colloidal systems.”

2. The paragraph after Eq.(3) the authors appeal to energy conservation. However, this is a viscous fluid in contact with its surroundings so energy conservation is not strictly followed here. I think the process being described here, though, happens fairly quickly compared to the rate at which heat can flow out so it is the separation of time scales for these processes that makes the authors argument valid here. A rephrasing, or footnote, could be useful here to clarify the argument.

We thank the Reviewer for pointing this out. We added a note to clarify this point in the revised manuscript.

In the revised manuscript, we added on page 8:

“By energy conservation, the rate of the energy gained by the tactoids due to the normal stresses from the flow field must be equal to the rate of the energy changes in the free energy of the tactoids associated with their elastic/interfacial energy. Note that setting the two energies equal stands valid here since the process happens significantly faster than the rate at which the heat can flow out, thus separating the time scale for the transfer of the energy associated with structural changes from the time scale for heat dissipation.”

3. I believe you are allowed 70 references in the main article where you have only 50. You should move 20 of the references in the supplementary material to the main references. (This ensures they are picked up by citation indexes which makes your article easier to find in a "forward" reference search so benefits you as well as the authors of these articles). We agree with the Reviewer. In the revised manuscript, we moved the references in the supplementary material (Refs. 5, 8, 12, 14-17, 19-24: that are missed to be cited in the main text previously) to the main article.

Reviewer #2

We would like to thank the Reviewer #2 for his/her comments on our manuscript. Below is our response with references to the pages of the revised manuscript.

By experimental work combined with numerical modeling and simulations, the authors investigate the relaxation of tactoids that were brought out of equilibrium by subjecting them to an elongational flow. The initially elongated nematic liquid crystalline droplets relax both in terms of shape (eventually assuming the characteristic lens-shaped form) as well as, under particular conditions, in terms of internal structure, where, from homogenous alignment, bipolar or cholesteric configurations can emerge. The manuscript reports that the shape relaxation follows an exponential decay (the experimental results being supported by the numerical results). For the structural relaxation the authors find that a first order decay properly describes two of the cases (i.e. where the equilibrium configuration is either homogenous or bipolar), whereas for the cholesteric equilibrium tactoids a second order exponential decay is necessary. The numerical simulation nicely agrees with the experimental finding only for the first two cases.

We would like to thank the Reviewer for this remark, helping us to clarify our message in the revised manuscript. In the revised manuscript, to highlight the degree of the agreement between the experimental results and numerical simulation in the third case, cholesteric tactoids, we added the rate of the changes in S (obtained using the equation of the fitted line) versus the scaled time obtained from numerical simulation in the revised Fig. 4f and compared it with experimental results. Now, it is clear that the numerical simulation also captures the second-order exponential decay observed experimentally for cholesteric tactoids structural relaxation. We additionally noted in the main text that while there is such a good agreement in capturing qualitatively the structural relaxation between the experiments and numerical simulations, the numerical simulation underestimates the exact values of the equilibrium order parameter obtained from experiments in the case of cholesteric tactoids.

In the revised manuscript, we modified Fig. 4c and f and the caption (see the *Italic text*). We additionally added a text to the revised manuscript as the following:

“

Fig. 4 | Structural relaxation of different classes of liquid crystalline tactoids. **a-c**, Evaluation of \mathcal{S} (defined as $\frac{S(t)-S_{\text{equil}}}{S_{\text{init}}-S_{\text{equil}}}$ for homogeneous and bipolar tactoids and $\mathcal{S} = \frac{S(t)-S_{\text{minimum}}}{S_{\text{init}}-S_{\text{minimum}}}$ for cholesteric tactoids) with respect to scaled time, $\frac{t}{\tau_c}$ with τ_c the characteristic configurational relaxation time, for tactoids that relax to homogenous (**a**), bipolar (**b**) and cholesteric (**c**) configurations at equilibrium. The experimental insets showing the retardance images taken with LC-PolScope along with numerical simulation results present the critical state of the relaxation for each class of the tactoids. Color bar denotes the order parameter values (S) in numerical simulation insets. Note that the brightness of the experimental images is increased for better visualization. The symbols denote the experimental data, *black solid lines are numerical simulation results*. *Colored and dashed black lines show the fitting that is used to obtain τ_c from experimental and numerical simulation results, respectively*. **d-f**, The changes in $\frac{dS}{dt}$, obtained from the fitted lines in (**a-c**), during relaxation for different classes of tactoids: homogenous (**d**), bipolar (**e**) and cholesteric (**f**) configurations. While homogenous and bipolar tactoids follow monotonic single exponential decay during relaxation $\mathcal{S} = \exp\left(-\frac{t}{\tau_c}\right)$, the cholesteric tactoids are characterized by a non-monotonic behavior of \mathcal{S} during relaxation (see panel (**c**)), well described by a second order exponential decay, $\mathcal{S} = c_1 \exp\left(-t/\tau_{c,1}\right) + (1 - c_1)\exp\left(-t/\tau_{c,2}\right)$, where c_1 is a constant.”

On page 10 in the revised manuscript, we added:

“It is clear from Fig. 4 that numerical simulations capture the first order exponential decay in the case of homogenous and bipolar tactoids. In the case of cholesteric tactoids, our numerical simulation results bear qualitatively similar behavior for the structural relaxation of the tactoids although simulations underestimate the equilibrium order parameter obtained experimentally for cholesteric tactoids structural relaxation (Fig. 4c). To illustrate this further, we show the rate of the changes in \mathcal{S} versus the scaled time in Figure 4d-f.”

The submitted manuscript is a follow-up of the recent paper by part of the authors of the current work (ref 9, published as well in Nature Communications). In both works (i.e. this manuscript and ref 9), the behavior of amyloid fibril tactoids under flow and their relaxation is investigated using

a microfluidic setup. In ref 9, the focus is set on the shape changes and internal order transitions of the tactoids, whereas here the emphasis lays more on the relaxation mechanism. The current work features as well a numerical modeling of the relaxation. Another appealing addition compared to the previously published work is the two experimental systems (a different preparation of amyloid fibrils and a cellulose based liquid crystalline material) which gives more generality to the work.

From the perspective of the topics tackled in this manuscript, this work is more specialized than the previous paper (ref 9) and therefore less susceptible for a larger audience. Even if the paper is generally well written, the introduction does not give a convincing picture of why the topic investigated here is a crucial step forward in liquid crystal physics or what the implications or uses of the findings in this manuscript are. A more comprehensive range of applications would have probably strengthened the claim of the authors.

As the Reviewer pointed, in the revised manuscript, we modified the Introduction to deliver clearly why the topic investigated here is important. We added a comprehensive range of the applications to which our work is relevant, and mentioned that our study has implications in the fields of active nematic and biological processes where understanding of the hydrodynamics and relaxation of the liquid crystalline systems can be of great benefit, see the following.

We modified on page 2:

“Colloidal liquid crystals are a class of soft matter formed when shape-anisotropic nanoparticles are dispersed in a an isotropic fluid¹. When confined to a finite volume, rod-like colloidal particles self-organize into various structures that are set by a delicate balance between anisotropic viscoelastic and surface properties²⁻⁴. The subtle balance between these contributions results in facile response to external fields^{1,5} such as mechanical, flow, electric, and magnetic, giving rise to many opportunities and technological applications. Example includes displays, spatial light modulator and tunable filters in medical devices and optics, liquid crystal biosensors for rapid diagnostics, and new functional material such as artificial muscles exploiting liquid crystalline anisotropic physical properties⁶⁻⁸. However, facile responsiveness to external fields (and disturbance) makes them very fragile to be studied experimentally under dynamical conditions^{5,9}. In particular, relaxation of liquid crystalline droplets under mobile confinement is still poorly understood despite its central importance in a variety of phenomena in condensed matter physics. This includes particle packing¹⁰, self-assembly¹¹, and relaxation of colloidal

liquids¹² with implications in the field of active nematic, e.g. living liquid crystals¹³, where the understanding of the hydrodynamics of the liquid crystals is critical¹⁴.

Liquid crystalline droplets, known as tactoids, are a particularly significant example of colloidal liquid crystals, since they consist of micro-confined liquid crystalline colloids with a self-selected shape/structure stemming out from the thermodynamically-driven phenomena from which they emerge, i.e. spontaneous liquid-liquid crystalline phase separation (LLCPS)¹⁵⁻²³. In stark contrast with spherical liquid crystalline emulsions, achieved commonly by emulsifying liquid crystals in another immiscible liquid (like water in oil)²⁴⁻²⁵, tactoids hold spindle-like, prolate, or oblate shapes with different nematic-cholesteric internal structures^{2-4,15-22,26}, as a consequence of the coupling between the vanishingly small interfacial tension, the surface anchoring at the interface, the chirality of colloids and the anisotropic elastic properties^{3,4}. These features make tactoids a very unique system with peculiar viscoelastic²⁷⁻²⁹ and boundary^{3,4} properties, thus adding theoretical challenges to the experimental ones when describing these complex colloidal systems under dynamical conditions. For instance, recent experiments suggest that the boundary has a significant impact on the local structure of colloids³⁰⁻³³ and on the equilibration pathways of structural relaxation of colloidal systems³⁴; yet such understanding mainly comes from the examination of colloidal systems with static boundary conditions³⁰⁻³⁵. Moreover, one of the challenges of the current study is to disentangle the rate of self-assembly of the liquid crystalline tactoids from shape relaxation, providing insights on the kinetics of self-assembled complex colloidal systems.”

One aspect that needs to be clarified concerns the breakage and merging of tactoids. These two types of events are briefly mentioned in the previous publication (ref 9), but are completely absent from this manuscript. I would expect that tactoids that have resulted from, for example, a coalescence event would have quite a different equilibration dynamic both in terms of shape as well as structure and therefore I guess the authors choose not to present the corresponding data in this manuscript. What is the probability of occurrence of a breakage of a tactoid in two or more smaller size components or two tactoids merging?

We must note up-front that events such as break-up and coalescence are not (and cannot be) studied in this work for a precise reason: they imply volume changes, and hence they lead to a switch in the internal symmetry of the liquid crystalline director field, as discussed, for example, in reference 4 and 19. In other words, the present work relies on the important overarching point that $Volume_{rest} =$

$Volume_{stretched} = Volume_{relaxed}$. If events such as break-up or coalescence were to be included, the nematic field director configuration would change not only due to rheological perturbations, but also due to volume changes (for example: two small volume bipolar droplets may change to a larger volume cholesteric droplet upon coalescence: see reference 4).

This being said, a number of points may be noted with respect to break-up and merging, as discussed below.

We did occasionally observe the merging of the two tactoids mainly when two tactoids approach at the same time to the elongation region (see for example two tactoids in the dashed box in the following figure). Thus, the probability of occurrence of the two tactoids merging in this system depends on how crowded of tactoids the system is. The number of the tactoids inside the chip is in direct link with the concentration of the solution within the co-existence region of the isotropic and nematic. The higher the concentration, the higher the number of the tactoids that are formed, thus the higher the number of the merging event. However, as in our study where we study the relaxation of the tactoids individually and at constant volume, we paid attention to stay at very low crowdedness in the system to make sure that that tactoid behavior is not affected by the presence of other tactoids.

Fig. 1 | Merging of the tactoids in the elongation region. Two tactoids merge when they approach the elongation region at the same time, highlighted by dashed box (a). The merging point on the tactoids differ from each other (b)-(d).

We should also note by two additional points that the above set-up cannot provide quantitative understanding of the merging tactoids which the Reviewer pointed in the present and next remark. First, there is no control on the merging point (whether for instance it is head-to-head merging or off-center), see the above figure panel b-d. Second, the experiment of the merging here is totally different from what is known or studied before as merging of the tactoids. This is so as the tactoids here merge when they are already stretched and in homogenous configuration and they keep stretching further in homogenous configuration (see figure above).

In term of the breakup, we observed the breakup in the relaxation region of the channel (see the following figure), however in very few tactoids.

Supplementary Fig. 1 | Schematic of microfluidic system contraction-abrupt expansions geometry used to study the elongation and relaxation of the tactoids. A liquid crystalline suspension with a concentration within the isotropic–nematic coexistence region is injected to the microfluidic system, allowing to form tactoids with various volumes at upstream region. Tactoids travel in flow direction and are elongated in the elongation region in the flow direction. Right at the beginning of the relaxation region, tactoids are elongated perpendicularly with respect to the flow direction and undergo relaxation. The coordinate x - y - z is located on the center line of the channel at the beginning of contraction zone. The coordinate r' - z' - θ' is located at the center of the tactoid. The geometry of the contraction zone is defined as $w(x) = x_1/(x_2+x)$, where $x_1 = l_e w_e w_i / [(w_e - w_i)]$, $x_2 = l_e w_i / (w_e - w_i)$, $w(x)$ is the width of the elongation region, l_e is the length of extension region, l_e is the length of elongation region, l_u is the length of upstream, h is the height of the channel, w_e is the upstream or downstream (extension region) width, and w_i is the throat width.”

(Note that the above figure is now added to the revised SI on page 14)

Detailed study is needed to assess the dynamics of the breakup and the condition in which the instability grows on the surface of the stretched tactoids, leading to breakup. In our study, we focus on the relaxation of the tactoids while their volume remains the same, while merging and breakup involves the changes in tactoids volume. To push our analysis further, following the remarks from the Reviewer, within the scope of our study, we additionally performed the analysis on the shape relaxation of the tactoids, that individually undergo shape relaxation towards equilibrium *after* the breakup so the volume of the tactoid stays the same (see the following part that is added to the revised manuscript). We observed that the shape relaxation of the tactoids, resulted from the breakup event, follows -as expected- single exponential decay, similar to the initially extended tactoids relaxation behavior as reported in the main text.

In the revised SI on page 14, we added the following Figure:

“

Supplementary Fig. 2 | Shape relaxation of amyloid fibril liquid crystalline tactoids following the breakup event. **a**, The LC (liquid crystal)-PolScope images capturing the breakup of the tactoids that is followed by relaxation of the resultant two tactoids. The time zero here is defined as the moment when the two resultant tactoids are disconnected. **b**, Experimental results, showing that shape relaxation of the tactoids, resulted from the breakup event, follow single exponential decay similar to the initially extended tactoids relaxation.”

We mentioned the above results in the main revised manuscript on page 5 as the following:

“We additionally performed experiments on the shape relaxation of the tactoids resulting from breakup events of initially extended tactoids in homogeneous configuration, and observed a single exponential decay trend similar to the initially extended tactoids relaxation behavior, see Supplementary Fig. 3.”

Furthermore, coalescing of nematic liquid crystalline droplets often results in tactoids containing topological defects that arise from the mismatch between the directions of alignment of the merging tactoids. Two examples out of the literature dealing with this problem are the following:

Kim, Young-Ki, Sergij V. Shiyankovskii, and Oleg D. Lavrentovich. "Morphogenesis of defects and tactoids during isotropic–nematic phase transition in self-assembled lyotropic chromonic liquid crystals." *Journal of Physics: Condensed Matter* 25, no. 40 (2013): 404202.

Zhang, Chiqun, Amit Acharya, Noel J. Walkington, and Oleg D. Lavrentovich. "Computational modelling of tactoid dynamics in chromonic liquid crystals." *Liquid Crystals* 45, no. 7 (2018): 1084-1100.

Have the authors observed such a behavior in their experiments? If so, could the authors comment on the equilibration of such defects-containing tactoids?

As we mentioned in our reply to the previous remark, the experiment of the merging here is totally different from what is known or studied before. This is so as the tactoids here merge when they are already stretched, both in homogenous configuration, with the main geometrical axis parallel to the flow direction and the homogeneous nematic field, and keeping stretching while the merging event continues. In other words, there is no “*mismatch between the directions of alignment of the merging tactoids*” as the reviewer is suggesting. Thus, within our experimental setup, the typical defects that are reported by simulation for the merging of the tactoids are neither observed (at least with the available experimental setup), nor relevant.

The modeling presented in the manuscript cannot capture, at least in the present form, events like breaking or merging of tactoids. Would it be possible to extended it to capture such events? Could the authors comment on this aspect?

Despite not being relevant to the problem treated in the present manuscript for the reasons discussed above, we have shown in our previous works (see following references) that our approach can result in high fidelity simulations capturing the dynamics of spatio-temporal liquid crystalline self-assembly including breakage, coalescence, and defect evolution. Hence, the modeling or simulation used in the present study can unequivocally capture breakage, merging, and defects.

1. Khadem, S. A., & Rey, A. D. Nucleation and growth of cholesteric collagen tactoids: A time-series statistical analysis based on integration of direct numerical simulation (DNS) and long short-term memory recurrent neural network (LSTM-RNN). *Journal of Colloid and Interface Science* **582**, 859-873 (2021).
2. Khadem, S. A. & Rey, A. D. Theoretical platform for liquid-crystalline self-assembly of collagen-based biomaterials. *Front. Phys.* **7**, 88 (2019).

However, we would like to note that, as we described in the reply to the previous remarks, providing analysis on the breakage and coalescence of the tactoids, is out of the scope of present study. In our study we focus on the relaxation of the tactoids while the volume of the tactoids remain invariant. While the merging and breakup involves a change in the volume of the tactoids.

To address the above in the revised manuscript, we edited our text on page 14 as the following:

“Direct Numerical Simulations. Past studies^{23,28,56-58} have shown that the time-dependent Ginzburg-Landau model can capture the spatio-temporal coupled relaxation dynamics of shape and structure. Furthermore, our approach can result in high fidelity simulations capturing dynamics of spatio-temporal liquid crystalline self-assembly including breakage, coalescence, and defect evolution. However, these are out of scope of the current study. In the present study, we applied this modeling approach and we focused on the relaxation dynamics of the shape and structure in initially extended

tactoids, which are isolated in isotropic phase. The implementation has been elaborated in detail in our previous works—See Ref. 23, 28 and 56.”

The relaxation of tactoids has been also already investigated, however in a different set-up, where confinement is pre-imposed in the following paper:

Gârlea, Ioana C., Oliver Dammone, José Alvarado, Valerie Notenboom, Yunfei Jia, Gijse H. Koenderink, Dirk GAL Aarts, M. Paul Lettinga, and Bela M. Mulder. "Colloidal liquid crystals confined to synthetic tactoids." *Scientific reports* 9, no. 1 (2019): 1-11.

There, the authors report a variety of different equilibration pathways which result from a mismatch between initial orientation of the liquid crystal and confinement-imposed orientation. Could the authors comment on the results in this paper in connection with their findings?

We find the paper highlighted by the Reviewer a very interesting reading, albeit with little relevance to the study presented in our manuscript. To start with, the work is limited to **2D geometries**, in contrast to the **3D geometry** considered in our work; secondly, the reference suggested provides no information about the relaxation kinetics of the tactoids, which is the main problem treated in our work. Additionally, that work addresses 2D structures **only by simulations** and under **static** confinements, that is under conditions very far from those used in our work. As such, it is difficult to find a connection between that reference and our work. What the reference suggested by the Reviewer does very well, however, is to highlight the importance of this topic and the obstacle that are present to study experimentally and through simulations the relaxation of the tactoids.

Precisely that same paper (see the following), supports our statements in the abstract that *the relaxation mechanisms for liquid crystalline colloids under mobile boundaries remain still unexplored* and, that in our study, for the first time, we present experimentally the relaxation of liquid crystalline tactoids in 3D, supported by the numerical simulations, where both the shape and structure of the tactoids undergo relaxation.

Indeed, in such a reference the following text from the last paragraph can be read:

“Finally, while it is interesting to speculate about the behavior of colloidal liquid crystals in true 3D confinement, there are at present serious practical obstacles to doing so. On the experimental side, creating such cavities with the current state-of-the-art soft lithography techniques employed is unfortunately not possible. Although there are novel techniques on the horizon that could make fully 3D shapes realizable in glass sometime in the future...”

More importantly, in the above paper, while different pathways of the relaxation of the structures are reported by the simulation, the initial director field condition where all particles are initially aligned to the main axis of the tactoid, as to what we observe in the initially extended tactoids, was never studied. We understood the reason from the paper as the following (the text is from the paper):

“...This would require a computationally very expensive analysis with high statistics of the local orientational ordering fields to obtain a reliable marker for the transition. We chose not to dwell on this issue here.”

Thus, we believe that while the mentioned paper nicely present advances on the structure of the liquid crystalline systems in the 2D static confinement, it does not provide information on both shape and structural relaxation of the initially extended tactoids, which makes comparison with our present work difficult.

In summary, due to the above mentioned differences, we are limited to comment directly on the results of the mentioned paper in connection with our findings, but we now cite the above paper when talking about the effect of the boundary condition on the structural relaxation of the colloidal systems on page 2 and edited the text to reflect the above-mentioned paper (appears as Ref. 34 in the revised manuscript):

“For instance, recent experiments suggest that the boundary has a significant impact on the local structure of colloids³⁰⁻³³ and on the equilibration pathways of structural relaxation of colloidal systems³⁴; yet such understanding mainly comes from the examination of colloidal systems with static boundary conditions³⁰⁻³⁵.”

Considering the above remarks, I would not recommend publications in Nature Communication (at least not in the current form).

We believe that the actions taken with respect to all the remarks of the Reviewer, and the improvements made to the manuscript, should make more obvious the significance of the work presented and why it should be a good fit for Nature Communications.

Minor remark:

Figures are sometimes hard to read due to crowding of elements, hindering the reader to properly see the results. One example is Figure 1, where, in the Simulation panels, in addition to color-coded director field orientation (in a particular plane), the orientation given by small red lines is shown. Due to the very dense mesh, it is: firstly, impossible to properly distinguish the orientation of these lines (at least in a printed size of the figure) since they are much too small and, secondly, these lines partially hide the director field below. I suggest decreasing the density of lines which would allow for size increase or removing them completely (for panel a and b they do not provide additional information).

We modified Figure 1 to make it easier to read. As suggested by the Reviewer, in the simulation panels:

- a) Figure 1a and b, we removed the red lines allowing to clearly show the director field below. As it is exactly pointed out by the Reviewer, the red lines in panel a and b do not provide additional information as due to the axisymmetric nature of the homogenous (panel a) and bipolar (panel b) tactoids, colors and lines show the same information. We added a text to the caption of the figure 1 to highlight this point, see italic text below in the Fig. 1 caption.
- b) Figure 1c, we decreased density of lines and increased size of lines.

In the revised manuscript on page 22, we improved Figure 1 and its caption as following.

Fig. 1 | Relaxation of different classes of the β -lactoglobulin amyloid fibril liquid crystalline droplets. The sequence of time-lapse images of relaxation of initially extended amyloid fibrils liquid crystalline droplets with different volumes. In each panel, the first row shows the experimental results taken with LC (liquid crystal)-PolScope device. The colormaps corresponding to experimental results denote the orientation of the director field in the x - z plane; the second row demonstrates the numerical simulation results with color bar capturing the director field orientation with respect to z -axis. The tactoids are at the homogenous configuration at the initial state and upon relaxation, they hold different configurations. **a**, Initially extended tactoid with volume $644 \mu\text{m}^3$ undergoes shape relaxation while its configuration remains unchanged at homogenous configuration. **b**, An elongated tactoid with volume $2751 \mu\text{m}^3$ relaxes both its shape and structure recovering a bipolar configuration upon relaxation. **c**, A droplet with volume $16414 \mu\text{m}^3$, having larger volumes compared to **(a)** and **(b)**, relaxes to a cholesteric structure with three bands. *Since colors and director lines show the same information due to the axisymmetric nature of the*

homogenous (a) and bipolar (b) tactoids, the lines are not shown for better readability (see Supplementary Videos 4-6 for the version with lines). Note that the brightness of the experimental images is varied for better visualization.

Another example is Figure 2, panel e, where a number of very similar markers both in shape and color are used. A different choice of symbols/colors would help.

We agree with the Reviewer. In the revised manuscript, we changed the colors/symbols in Figure 2, panel e, to make it easier to distinguish different data points.

We improved Figure 2 on page 23 as following:

“

Fig. 2 | Shape relaxation of amyloid fibril liquid crystalline tactoids. a-c, Evaluation of \mathcal{R} (defined as $\frac{R(t)-R_{\text{equiv.}}}{R_{\text{init.}}-R_{\text{equiv.}}}$) with respect to time, t , for tactoids that relax to homogenous, $R_{\text{equiv.}} = 9.4 \mu\text{m}$ (a), bipolar, $R_{\text{equiv.}} = 19.3 \mu\text{m}$ (b) and cholesteric, $R_{\text{equiv.}} = 27.8 \mu\text{m}$ (c) configurations at equilibrium. Symbols and black lines denote the experimental and numerical simulation results, respectively; colored lines show the fitting ($\mathcal{R} = \exp(-\frac{t}{\tau_s})$) that is used to obtain the characteristic shape relaxation time, τ_s . d, Evaluation of \mathcal{R} with respect to scaled time t/τ_s resulting in a universal curve, $\mathcal{R} = \exp(-\frac{t}{\tau_s})$, for shape relaxation of the different classes of tactoids with various volumes and initial elongation values. e, Circle, triangle, and square symbols denote homogenous, bipolar, and cholesteric tactoids, respectively. The filled, half-filled, and empty symbols, respectively, correspond to BLG I, BLG II, and SCNC liquid crystals, see Table 1. The error bars represent standard deviation. The developed theory, solid line, predicts the τ_s for different classes of BLG and SCNC liquid crystalline tactoids, confirming the generality of our approach to predict the bio-colloidal liquid crystalline tactoids relaxation behavior.”

Reviewer #3

We would like to thank Reviewer #3 for his/her comments on our manuscript. Below is our response with references to the pages of the revised manuscript.

The authors study the relaxation of the shape and liquid crystal structure of externally elongated tactoids towards equilibrium (both experimentally and numerically). They show that for different volumes of the tactoids, the shape relaxation is followed by an exponential decay towards the equilibrium. The time scale of this decay has isotropic (similar to simple droplets) and anisotropic (due to the liquid crystalline structure) contributions.

The relaxation of the nematic order parameter, however, has first and second-order exponential decays, depending on the ground state of the liquid crystal.

In addition to experiments, the following theoretical approaches are used to understand the experiments:

1-The time scale of the shape relaxation is found using dimensional analysis.

2- Using energy conservation, the authors find the initial shape of the tactoid due to the external flow.

3- Finally, numerical simulations have been used to study the evolution of the liquid crystal.

The question that has been tried to answer is interesting, however, there are some points that are very unclear to me (see the comments below). If the authors can precisely address all these points, I would be happy to recommend the manuscript for publication.

1- In eq. 1 for the time scale of the shape relaxation, if bend/splay elastic coefficient becomes larger than the twist elastic coefficient ($2K < K_2$) the time scale becomes imaginary. Can the authors comment on this? Also, due to the presence of the term $(2K/K_2 - 1)^{0.5}$ in the denominator, the timescale can diverge when $2K/K_2$ goes to 1. Could the authors please clarify what this means?

We would like to thank the Reviewer for raising this constructive point. We should have mentioned that K/K_2 is always greater than $1/2$, thus $2K/K_2 - 1$ is greater than 0. This is supported both theoretically and experimentally. Theoretically, it is obtained by Odijk (Ref. 1 in the following) that if the rods in the liquid crystalline system are rigid the excluded volume theory predicts $K/K_2 = 3$. Additionally, K/K_2 is greater than $1/2$, as derived by Ericksen² by discussing the conditions keeping free energy positively-definite. Ericksen derived that $\frac{K}{K_2 + K_{24}} \geq 1/2$, which considering that $K_{24} > 0$, gives $\frac{K}{K_2} > 1/2$. Experimentally, we can refer to the Table 1 in Reference 3 (in the following) which list several lyotropic rod-shaped liquid crystals and a thermotropic liquid crystals all with K/K_2 higher than 2. To add more

experimental support, we can refer to the References 4 and 5 (in the following) where K/K_2 is measured to be higher than $1/2$.

We improved the manuscript by including this point in the revised manuscript as the following:

On page 6 we added:

“It should be noted that, from both theory¹⁻² and experimental measurements³⁻⁵ on different systems of rigid rod-shaped liquid crystals, including filamentous colloids⁶ analogous to those studies here, the ratio of $\frac{K}{K_2}$ in Eq. 1 is always greater than $1/2$, and thus Eq.1 only contains real arguments.”

1. Odijk, T. Elastic constants of nematic solutions of rod-like and semi-flexible polymers. *Liq. Cryst.* **1**, 553–559 (1986).
2. Ericksen, J. L. Inequalities in liquid crystal theory. *Phys. Fluids* **9**, 1205 (1966).
3. Dietrich, C. F., Collings, P. J., Sottmann, T., Rudquist, P. & Giesselmann, F. Extremely small twist elastic constants in lyotropic nematic liquid crystals. *Proc. Natl Acad. Sci. USA* **117**, 27238-27244 (2020).
4. Zhou, S. et al. Elasticity of lyotropic chromonic liquid crystals probed by director reorientation in a magnetic field. *Phys. Rev. Lett.* **109**, 037801 (2012).
5. Taratuta, V. G., Hurd, A. J. & Meyer, R. B. Light-scattering study of a polymer nematic liquid crystal. *Phys. Rev. Lett.* **55**, 246 (1985).
6. Bagnani, M., Azzari, P., De Michele, C., Arcari, M. & Mezzenga, R. Elastic constants of biological filamentous colloids: estimation and implications on nematic and cholesteric tactoid morphologies. *Soft Matter* **17**, 2158-2169 (2021).

2- I suggest that the authors compare the experimental data on the initial shape of the tactoid with the solution of eq. 4 (adding data points to Fig. 3). Without a comparison, equation 4 is a mathematical work-out that has not been used throughout the paper.

As the Reviewer suggested, we performed the experiments of the elongation of the tactoids under different extension rate; see below. We would like to thank the Reviewer for pointing this out, allowing us improving the manuscript significantly. Comparing the prediction of the Eq. 4 and the experimental results, we find a good qualitative and quantitative agreement, given that the theoretical estimates are based on scaling concepts! To evaluate the quantitative agreement, we present the data with only one pre-factor, c , in the equation 4 as the following:

$$5\gamma\omega r^6 + 6c\mu_1\dot{\epsilon}V^2r - \gamma V^2 = 0$$

where c is 0.14. This pre-factor is entirely justified since the analysis is based on the scaling form of the Frank-Oseen energy landscape, in fact, scaling laws would allow using more than one pre-factor

considering the different nature of the scaling terms of the Frank-Oseen energy landscape (first and the last terms in the above equation). It is therefore remarkable that good agreement is reached with only one pre-factor in the scaling terms. Additionally, we should note that the experiments of the large volume of the tactoids (e.g. $V=30,000 \mu\text{m}^3$) at the high extension rate is limited by the experimental setup. This is so as the length of the tactoids in the stretched forms becomes extremely high compared to the size of taken images, even using the microscope at low magnifications e.g. objective 5x with a field of view of $1.8 \times 1.3 \text{ mm}$, preventing to fully capture tactoids with large volume at very high extension rate. The experimental points are those observable within the possibilities offered by our setup.

We added the revised Fig. 3 on page 24 in the revised manuscript and modified the text as the following:

“

Fig. 3 | Tactoids deformation under extensional flow field predicting the maximum initial deformation of tactoids under various extension rate. a, The theory (lines) and the experimental data (symbols) predict that short axis of the tactoids r decreases as the extension rate increases where r lines, corresponding to tactoids with different volumes, converge to a single curve at large values of extension rate. **b,** At a given extension rate, the short axis of the tactoids increases logarithmically with an increase in the volume of the tactoids.”

We modified the text following equation 4 on page 8 as the following:

$$5\gamma\omega r^6 + 6\mu_I \dot{\epsilon} V^2 r - \gamma V^2 = 0, \quad (4)$$

giving the steady-state elongated shape of the tactoids under a given extensional flow field in terms of r as a function of $\dot{\epsilon}$ and V . There is no analytical solution for equation 4, but the numerical solution along with our experimental data are presented in Figure 3. Here, to best match experimental observations, the second term is re-scaled by a pre-factor of 0.14, which is fully justified by the use of a scaling form of the Frank-Oseen energy landscape. Our results suggest that the short axis of the tactoids r decreases as the extension rate increases, where r lines, corresponding to tactoids with different volumes, converge to a single universal curve at large values of extension rate (Fig. 3a). The most remarkable consequence

of our analysis is that for high extension rates, r becomes independent of the volume, that is, the cross section of the tactoid is simply ruled by extension rate, and that at identical extension rates, tactoids of different volumes V only differ by their long radius R , which is directly proportional to V . In the regime of low extension rate, the short axis of the tactoids becomes volume-dependent and increases logarithmically with the increase in the volume of the tactoids (Fig. 3b), which is nicely supported by the collected experimental dataset. Equation 4 can be used to predict the $R_{init.}(=V/r^2)$ as the maximum tactoids deformation that can be reached under a given extensional flow rate. Note that in Figure 3 the experimental data of the tactoids with large volumes at high extension rate are absent, as this regime corresponds to elongated tactoids stretching beyond the field of view of the optical microscope images.”

3- Page 9, last paragraph:

It is clear from the experimental data there are two time-scales. But more explanation is needed here. Does the above paragraph mean that the system first relaxes to a nematic phase and then the cholesteric phase starts to appear? If so, why does the system first go to a nematic phase instead of directly going to a cholesteric phase?

Also, what do they mean by “bend/splay and twist occur at different length-scale”? Why are these length scales different?

We would like to mention that on page 9, as the Reviewer pointed, first we explained that there are two relaxation time-scales for the cholesteric tactoids. We supported this argument in the manuscript “when tactoids relax to homogenous and bipolar tactoids, they undergo bending and splay relaxation; when the configuration relaxes to cholesteric tactoids, an additional twist relaxation takes place (see numerical simulation results in Supplementary Videos 3-6). This suggests that the second exponential decay associated with the cholesteric tactoids originates from the twist term.”

Next, we suggest that in $\mathcal{S} = c_1 \exp(-t/\tau_{c,1}) + (1 - c_1)\exp(-t/\tau_{c,2})$, where $\tau_{c,2}$ is significantly longer than $\tau_{c,1}$, the second exponential decay $\tau_{c,2}$ originates from the twist term and $\tau_{c,1}$ originates from splay/bend. We base this statement on two grounds.

First, we compare the length-scales of the nematic and cholesteric ordering. We consider the length-scale of the cholesteric phase as the length which is required for the phase to form a single periodic twist, that is set by the inverse wave number (i.e. the pitch) which is in the order of 10^1 micrometers. To form nematic ordering (characterized by splay and bending), the length scale is defined in the range of the length of the fibrils (mesogens) which is in the order of 10^{-1} micrometers. Thus, we argue that the larger twist length-scales compared to splay/bend imply longer relaxation times for the twisting deformation.

Second, in reply to the Reviewer remark, we performed a new set of experiments of the relaxation of the cholesteric tactoids using a Polscope system allowing us to capture the changes in the director field over time and to follow the twist dynamics in the director field (see the following). From the director field data, it is clear that the change/rotation in the director field (showing the twist) takes place until the latest stages of the relaxation process and the twist changes are significant in the latest stage. While from the data in the following figure one could argue that the change in the director field is within the nematic phase first and only later the twist behavior appears, we refrain from claiming that the system first relaxes to a nematic and then the cholesteric phase, since we are limited from the experimental data to precisely when the nematic ordering or cholesteric ordering starts. Thus, in the revised manuscript, we mention that change/rotation in the director field (the twist) takes place until the end of the relaxation process and shows significant changes at the late stage, suggesting that the second and longer exponential decay $\tau_{c,2}$ originates from the twist.

In the revised manuscript, we modified the text on page 10 as the following:

“To interpret the physics behind these structural relaxations, we inspect the time scale related to bending, splay and twist terms in tactoids. Compared to the relaxation to homogeneous/bipolar tactoids, which involves nematic ordering with at best splay/bend relaxation, when the configuration relaxes to cholesteric tactoids, an additional twist relaxation takes place (see numerical simulation results in Supplementary Videos 4-6). This suggests that the second exponential decay associated with the cholesteric tactoids originates from the twist term. We suggest that in $S = c_1 \exp(-t/\tau_{c,1}) + (1 - c_1)\exp(-t/\tau_{c,2})$, where $\tau_{c,2}$ is significantly longer than $\tau_{c,1}$, the second exponential decay $\tau_{c,2}$ originates from the chiral twist term while $\tau_{c,1}$ originates from simple nematic ordering. We base this statement on two grounds. First, we compare the length-scales of the nematic and cholesteric ordering. We consider the length-scale of the cholesteric phase as the length which is required for the phase to form a single periodic twist, that is set by the inverse wave number (i.e. the pitch) which is in the order of 10^1 micrometers. In contrast, to form nematic ordering (characterized by splay and bending), the length scale is defined in the range of the length of the fibrils (mesogens) which is in the order of 10^{-1} micrometers. Thus, we argue that the larger twist length-scales compared to splay/bend imply longer relaxation times for the twisting deformation. Secondly, from the experiments of the relaxation of the cholesteric tactoids using a Polscope, allowing us to capture the changes in the director field over time and follow the twist dynamics in the director field (see SI, supplementary Figure S4), we find that change/rotation in the director field (the twist) takes place until the latest stages of the relaxation process and the twist changes are significant in the latest stage, suggesting again that the second exponential decay $\tau_{c,2}$ originates from the twist re-arrangement.”

In revised SI on page 16, we added:

“

Supplementary Fig. 5 | Shape and director field relaxation of the cholesteric tactoid. The LC (liquid crystal)-PolScope images showing the relaxation of the of amyloid fibril liquid crystalline tactoid. The director field data shows that the change/rotation in the director field extend to the end of the relaxation process and shows significant changes at the late stage.”

4- In Fig. 4, it would be helpful to show the two exponential fits for S in Part C of the figure.

We agree with the Reviewer and in the revised manuscript we show the two exponential fits for S in Fig. 4c.

In the revised manuscript the new Fig. 4 on page 25 has been improved as shown below. Additionally, we added a text (see italic text) to explain the added fitting in the caption.

Fig. 4 | Structural relaxation of different classes of liquid crystalline tactoids. **a-c**, Evaluation of \mathcal{S} (defined as $\frac{S(t)-S_{\text{equil.}}}{S_{\text{init.}}-S_{\text{equil.}}}$ for homogeneous and bipolar tactoids and $\mathcal{S} = \frac{S(t)-S_{\text{minimum}}}{S_{\text{init.}}-S_{\text{minimum}}}$ for cholesteric tactoids) with respect to scaled time, $\frac{t}{\tau_c}$ with τ_c the characteristic configurational relaxation time, for tactoids that relax to homogenous (**a**), bipolar (**b**) and cholesteric (**c**) configurations at equilibrium. The experimental insets showing the retardance images taken with LC-PolScope along with numerical simulation results present the critical state of the relaxation for each class of the tactoids. Color bar denotes the order parameter values (S) in numerical simulation insets. Note that the brightness of the experimental images is increased for better visualization. The symbols denote the experimental data, *black solid lines are numerical simulation results*. *Colored and dashed black lines show the fitting that is used to obtain τ_c from experimental and numerical simulation results, respectively*. **d-f**, The changes in $\frac{dS}{dt}$, obtained from the fitted lines in (**a-c**), during relaxation for different classes of tactoids: homogenous (**d**), bipolar (**e**) and cholesteric (**f**) configurations. While homogenous and bipolar tactoids follow monotonic single exponential decay during relaxation $\mathcal{S} = \exp\left(-\frac{t}{\tau_c}\right)$, the cholesteric tactoids are characterized by a non-monotonic behavior of \mathcal{S} during relaxation (see panel (**c**)), well described by a second order exponential decay, $\mathcal{S} = c_1 \exp\left(-t/\tau_{c,1}\right) + (1 - c_1)\exp\left(-t/\tau_{c,2}\right)$, where c_1 is a constant.”

5- The simulations of the evolution of the LC seem to fail to capture the decay towards the cholesteric phase (Fig. 4C). This has not been referred to throughout the text. If the model is generic enough, it should be able to capture the dynamics towards the cholesteric phase too.

We agree that this has not been duly discussed in the text. We would like to thank the Reviewer pointing this out, helping us to clarify this issue in the revised manuscript. In the revised manuscript, to highlight the degree of the agreement between the experimental results and numerical simulations in the case of cholesteric tactoids, we added the rate of the changes in \mathcal{S} (obtained using the equation of the fitted line) versus the scaled time obtained from numerical simulation in the revised Fig. 4f and compared it with the experimental results (see revised Fig. 4 in the reply to previous comment). Now, it is clear that numerical simulation also captures well the second-order exponential decay observed experimentally for cholesteric tactoids structural relaxation. We additionally noted in the main text that while there is

good agreement in capturing qualitatively the dynamics of the structural relaxation towards the cholesteric tactoids between the experiments and numerical simulation, the numerical simulation underestimates the exact values of the equilibrium order parameter obtained from experiments in the case of cholesteric tactoid.

On page 10 in the revised manuscript, we added:

“It is clear from Fig. 4 that numerical simulation nicely captures the first order exponential decay in the case of homogenous and bipolar tactoids. In the case of the cholesteric tactoids, our numerical simulation results bear qualitatively similar behavior for the structural relaxation of the tactoids although simulations underestimate the equilibrium order parameter obtained experimentally for cholesteric tactoids structural relaxation (Fig. 4c). To illustrate this further, we show the rate of the changes in \mathcal{S} versus the scaled time in Figure 4d-f.”

6- In experiments, is it possible to look at the evolution of the bend, splay, and twist distortions towards equilibrium? A graph with measurements in the elastic free energy of these deformations over time could be helpful in understanding the dynamics towards equilibrium.

We agree with the Reviewer that measurement of the elastic free energy of the tactoids deformation over time can provide helpful understanding on the dynamics toward equilibrium. However, this is strictly limited from the experimental side, as the measurements of the director field, which is needed to obtain the elastic contributes, are limited to 2D plane (say x-y plane). To the best of our knowledge, PolScope system (used in this study) is the most advanced system available to capture the director field information, yet this system still is limited to 2D (see the figure in our reply to the 3rd remark) and cannot capture director field information in 3D (x-y-z). The information in z direction is critical to measure experimentally elastic contributions, as one needs to calculate free energy landscape of the tactoid as following:

$$U_V \approx \int_V d^3r \left[\frac{1}{2} K [(\nabla \cdot n)^2 + |n \times (\nabla \times n)|^2] \right] + \frac{1}{2} K_2 (n \cdot \nabla \times n + q_\infty)^2 V$$

where, when it is expanded, the terms n_x , n_y , n_z , $\frac{\partial}{\partial x}$, $\frac{\partial}{\partial y}$, and $\frac{\partial}{\partial z}$ appear in all the above terms. With the experiments, we are only able to measure n_x , n_y , $\frac{\partial}{\partial x}$, and $\frac{\partial}{\partial y}$, but the information of n_z , and $\frac{\partial}{\partial z}$ cannot be captured. Thus, while we agree with the Reviewer that the pointed measurement is of importance, unfortunately from an experimental perspective this is not possible using the methodology used in the manuscript.

Minor comments:

a) Page 8, second line: both the second and the third terms have been ignored from Eq.

We thank the Reviewer for pointing this out, however the equation is correct: neither the twist nor the volume change in a continuously deforming homogeneous tactoid (i.e. under extensional flow). We added a sentence to the main text and SI to clarify this point.

In the revised main text on page 8 and in the revised SI on page 10, we added:

“Additionally, the third term is eliminated as, in the homogenous configuration and at constant tactoid volume, it does not change under deformation, so the rate of the energy gained by this term becomes zero.”

b) What is the final shape of the tactoid? Does it relax to a sphere or a spindle?

The final shape of the tactoid is the shape which the tactoid holds under equilibrium state. i.e. prior being subjected to the extensional flow. In other words, tactoids relax to their equilibrium shape which is either spindle-like or sphere-like shape depending on their initial volume. Essentially, the tactoids hold spindle shape at small volume size and by an increase in the volume, the shape changes toward spherical shape as the isotropic part of the surface free energy (γRr) in free energy landscape of the tactoid becomes dominant. For instance, cholesteric tactoids have large volumes thus usually hold aspect ratio just above 1. Homogenous tactoids are the smallest size of the observed tactoids whose aspect ratio can be estimated using Wulff construction as $2\omega^{1/2}$ for $\omega \geq 1$ and $\omega+1$ for $0 \leq \omega < 1$. The aspect ratio observed for the tactoids of the different solution is in the range of 1.5 to 4 (see following references 1-3 and Supplementary Fig. 5). Bipolar tactoids, with volume in-between homogenous and cholesteric tactoids, hold aspect ratio that is in between the one from homogenous and cholesteric tactoids, say around 1.5-2.5.

This point is very well understood based both on scaling forms of the Frank-Oseen energy landscape as well as its variational theory solution and has been discussed on several earlier works:

1. Jamali, V. et al. Experimental realization of crossover in shape and director field of nematic tactoids. *Phys. Rev. E* 91, 042507 (2015).
2. Bagnani, M., Azzari, P., De Michele, C., Arcari, M. & Mezzenga, R. Elastic constants of biological filamentous colloids: estimation and implications on nematic and cholesteric tactoid morphologies. *Soft Matter* 17, 2158-2169 (2021).

3. Nyström, G., Arcari, M. & Mezzenga, R. Confinement-induced liquid crystalline transitions in amyloid fibril cholesteric tactoids. *Nat. Nanotech.* **13**, 330 (2018).
4. Bagnani, M., Azzari, P., Assenza, S. & Mezzenga, R. Six-fold director field configuration in amyloid nematic and cholesteric phases. *Sci. Rep.* **9**, 1–9 (2019).

The final paragraph already explains how to predict the final morphology, however, following the reviewer comment, in the revised manuscript we edited the text on page 11 to clarify how to predict the final shape of the tactoid after relaxation, as the following:

“What is the configuration of the extended tactoid with a given initial volume after relaxation? We are able to predict the relaxed configuration of the tactoids using the theoretical modeling recently developed starting from a scaling form of Frank–Oseen elasticity theory⁴. According to this theory, the tactoids at equilibrium hold homogeneous configuration when $(V/\alpha)_{Homogenous} < (K/\gamma\omega)^3$, bipolar configuration when $(K/\gamma\omega)^3 < (V/\alpha)_{Bipolar} < [1.7\gamma/(K_2q_\infty^2)]^3$, and cholesteric configuration when $(V/\alpha)_{Cholesteric} > [1.7\gamma/(K_2q_\infty^2)]^3$. Approximating α equal to 3 for homogenous-bipolar and 1.5 for bipolar-cholesteric boundaries following Ref. 4, we computed these threshold values for BLG I and found $V_{Homogenous} \lesssim 800$, $800 \lesssim V_{Bipolar} \lesssim 11000$, and $V_{Cholesteric} \gtrsim 11000 \mu\text{m}^3$. This confirms that the tactoids shown in Figure 1 follow a relaxation path until equilibrium. Thus, knowing the initial volume of the tactoids, their configuration after relaxation can be predicted simply from the scaling form of Frank–Oseen elasticity theory and physical parameters of the system such as elastic constants, interfacial energy and anchoring strength⁴. We provide the nematic-cholesteric phase diagram of the tactoids collected from the samples in a cuvette at equilibrium showing tactoids configuration as a function of the volume in Supplementary Note 7, as a further demonstration that the initially stretched tactoids reach an equilibrium configuration after relaxation.”

c) It is not clear how the set-up in Fig. S3 is related to the cylindrical coordinate used in Eq. S16. I suggest that the authors use a 3D schematic figure to show the 3d experimental set-up. In addition, the region in which the drop elongates under the flow and the region where it relaxes is not clear. The expansion and compression regions are marked but it is not clear how the set-up works. A would suggest using a schematic time series of the different stages of the experiment.

In the revised manuscript, we added a 3D schematic figure showing the 3D experimental setup. As pointed by the Reviewer, we made sure that cylindrical coordinate, the region where the drop elongates under the flow and the region where the droplet relaxes are clearly shown. A times series of different stages of the experiments is schematically shown, see below.

Accordingly, we combined Figure S1 and S3 to newly improved Figure S1 in revised the SI file as following:

On page 14, in the revised SI file, it is added:

“

Supplementary Fig. 1 | Schematic of microfluidic system contraction-abrupt expansions geometry used to study the elongation and relaxation of the tactoids. A liquid crystalline suspension with a concentration within the isotropic–nematic coexistence region is injected to the microfluidic system, allowing to form tactoids with various volumes at upstream region. Tactoids travel in flow direction and are elongated in the elongation region in the flow direction. Right at the beginning of the relaxation region, tactoids are elongated perpendicularly with respect to the flow direction and undergo relaxation. The coordinate x - y - z is located on the center line of the channel at the beginning of contraction zone. The coordinate r' - z' - θ' is located at the center of the tactoid. The geometry of the contraction zone is defined as $w(x) = x_1/(x_2+x)$, where $x_1 = l_e w_e w_i / [(w_e - w_i)]$, $x_2 = l_e w_i / (w_e - w_i)$, $w(x)$ is the width of the elongation region, l_e is the length of extension region, l_c is the length of elongation region, l_u is the length of upstream, h is the height of the channel, w_e is the upstream or downstream (extension region) width, and w_i is the throat width.”

We additionally modified the text on page 2 in the revised SI to clearly explain the setup as the following:

“To perform the relaxation experiments of the tactoids, we use a microfluidic system with contraction-abrupt expansions design¹⁻², allowing to elongate the tactoids with different volumes and let them relax to the equilibrium (see Supplementary Figure 1). Tactoids with various volumes are formed inside the channel at upstream region. Tactoids travel in flow direction and are elongated as shown schematically in Supplementary Figure 1 in the elongation region (or contraction zone). Right after the elongation region, the tactoids are elongated in the perpendicular direction with respect to the flow direction and undergo relaxation to the equilibrium state in the relaxation region in Supplementary Figure 1. The

main reason for the elongation of the tactoids at the beginning of the relaxation region is the high extension rate of the flow in y -direction in the expansion zone, i.e. $\dot{\epsilon}_{yy} = \partial u_y / \partial y$, compared to the shear rate in the flow direction $\dot{\Gamma}_{xy} = \partial u_x / \partial y$. For the flow rate used in this study $U = 1.5 \mu\text{m s}^{-1}$ and our geometry, we found the ratio of the $\dot{\epsilon}_{yy} / |\dot{\Gamma}_{xy}|$ to be always over 40 along the centerline of the channel (for details we refer the interested readers to our recent study in Supplementary Ref. 3).”

d) In Eq. S23 why are the bend splay free energy proportional to area (r^2) and twist is proportional to volume (r^3)?

In Eq. S25 (where these terms appear - the Reviewer is referring as Eq. S23), the terms appear with the correct power law and with the correct dimensionality. Actually, while it is true that the twist term scales as the volume (r^3) as mentioned by the Reviewer, the splay+bend term scales as a length and not as an area: $r^2/R = r^2/(ra) = r/a$. All elastic contributes including bend, splay and twist free energy are proportional to the volume as reflected in the Frank-Oseen elasticity theory and adopted for the tactoids (Ref. 1 and 2 in the followings):

$$U_V \approx \int_V d^3r \left[\frac{1}{2} K [(\nabla \cdot n)^2 + |n \times (\nabla \times n)|^2] \right] + \frac{1}{2} K_2 (\theta + q_\infty)^2 V$$

In the above equation, to obtain a scaling estimate for the splay and bending, the free energy term of splay and bending is set to be proportional to K times V times the square of a reciprocal radius of curvature ($1/R$), meaning that the above equation scales as:

$$U_V \approx KV \left(\frac{1}{R} \right)^2 + \frac{1}{2} K_2 (\theta + q_\infty)^2 V$$

which then can be written (considering $V=Rr^2$) as Eq. 25 by following

$$U_V \approx \frac{Kr^2}{R} + \frac{1}{2} K_2 (\theta + q_\infty)^2 r^2 R$$

The first term (splay+bend) of this expression has been derived by Prinsen, P. & van der Schoot¹; the second has been derived by Nyström et al²:

3. Prinsen, P. & van der Schoot, P. Shape and director-field transformation of tactoids. *Phys. Rev. E* **68**, 21701 (2003).
4. Nyström, G., Arcari, M. & Mezzenga, R. Confinement-induced liquid crystalline transitions in amyloid fibril cholesteric tactoids. *Nat. Nanotech.* **13**, 330 (2018).

That this is correct can further be grasped by a quick dimensional analysis: $Y \sim [Nm^{-1}]$, $K, K_2 \sim [N]$, $(\theta + q_\infty)^2 \sim [m^{-2}]$, $\omega \sim [adimensional]$, $R, r \sim [m]$: all terms in eq. S25 scales as $[Nm]$, i.e. energy.

In the revised SI on page 10, to make this point clearer, we mentioned the reference by Nyström et al. when introducing Eq. S25, which provides the scaling for the full energy landscape inclusive of the twist term and wrote the Eq. S25 as the following:

“We now return to the free energy landscape of the tactoids, where the total free energy of the tactoid F_E is described in scaling form such as (Ref. 7):

$$F_E \sim \gamma R r \left[1 + \omega \left(\frac{r}{R} \right)^2 \right] + K V \left(\frac{1}{R} \right)^2 + \frac{1}{2} K_2 (\theta + q_\infty)^2 V \quad (S25)''$$

REVIEWERS' COMMENTS

Reviewer #1 (Remarks to the Author):

The authors made all the changes I suggested in my previous report. I recommend publication.

Reviewer #2 (Remarks to the Author):

In view of the additional work carried out, the clarifications and improvements made to the manuscript (especially to the introduction), and the authors' response to the comments, I find the work is suitable for publication.

Reviewer #3 (Remarks to the Author):

The authors have addressed my comments carefully and as such, I recommend publication of the current version. A few minor points:

1- Fig. 3a) in the reply: From the current data, it seems that the short axis of the tactoid (r) diverges for small extension rates. Surely, this is not the case. So adding additional data (maybe for zero extension rate where ' r ' is equal to its equilibrium value) could make it clear that the data saturates in small extension rates. Also, in part (b) of this figure, please add a comment on why there is no experimental data for large volumes for large extension rates.

2- Fig. 1 in the SI: "Right at the beginning of the relaxation region, tactoids are elongated perpendicularly with respect to the flow direction and undergo relaxation."

At the beginning of the relaxation region, do the tactoids get elongated again, or do the elongated tactoids from the extension region rotate? Please clarify this.

3-Fig. 4 part (b) please remove the error on the top of the graph.

Reviewer #1

The authors made all the changes I suggested in my previous report. I recommend publication.

We would like to thank the Reviewer for his/her insightful and fruitful remarks which helped us to improve our manuscript.

Reviewer #2

In view of the additional work carried out, the clarifications and improvements made to the manuscript (especially to the introduction), and the authors' response to the comments, I find the work is suitable for publication.

We would like to thank the Reviewer for his/her insightful and fruitful remarks which helped us to improve our manuscript.

Reviewer #3

The authors have addressed my comments carefully and as such, I recommend publication of the current version. A few minor points:

We would like to thank the Reviewer for his/her insightful and fruitful remarks which helped us to improve our manuscript. Below is our response to the minor points with references to the pages of the revised manuscript.

1- Fig. 3a) in the reply: From the current data, it seems that the short axis of the tactoid (r) diverges for small extension rates. Surely, this is not the case. So adding additional data (maybe for zero extension rate where ' r ' is equal to its equilibrium value) could make it clear that the data saturates in small extension rates. Also, in part (b) of this figure, please add a comment on why there is no experimental data for large volumes for large extension rates.

In response to first part of the Reviewer Remark that is related to Fig. 3a:

In the revised manuscript, as suggested by the Reviewer, we added additional data for zero extension rate as in the following. We note that we presented these data as new SI figure (Supplementary Fig. 4, see below) since the x axis in Fig. 3a is in log format (otherwise, in linear format, the trend of the data cannot be clearly seen) so that adding the additional data for zero value is impossible in the main text. In Supplementary Fig. 4, where x axis in linear form, we present the modeling and the experimental data for shear rate lower than 0.1 s^{-1} including zero extension rate. Additionally, we note that since the modeling (equation 4 in the main text) assumes the homogenous internal configuration for the tactoids under the shear rate, at zero extension rate we added only the data of the tactoids that hold homogenous configuration at equilibrium (or at zero shear extension rate).

In the revised SI on page 3, we added:

“Supplementary Fig. 4

Supplementary Fig. 4 | Maximum deformation of tactoids under various extension rate. The theory (lines) and the experimental data (symbols) predict the short axis of the tactoids r under various extension rate. The data corresponding to the zero shear where r is equal to its equilibrium value are obtained from the tactoids at equilibrium condition in a cuvette. Note that since the modeling (equation 4) assumes the homogenous internal configuration for the tactoids under the shear rate, at zero extension rate only the data of the tactoids that hold homogenous configuration at equilibrium (or at zero shear extension rate) are presented.”

In response to second part of the Reviewer Remark that is related to part (b):

We added a comment on page 9 why there is no experimental data for large volumes for large extension rates in the response to the Reviewer remark in the last revision. However, in the revised manuscript, we edited the text to make it clearer:

“Note that in Figure 3a-b the experiments of the large volume of the tactoids (e.g. $V = 30,000 \mu\text{m}^3$) at the high extension rate are limited by the experimental setup. This is so as the length of the tactoids in the stretched forms becomes extremely high compared to the size of taken images, preventing to fully capture tactoids with large volume at high extension rate.”

2- Fig. 1 in the SI: "Right at the beginning of the relaxation region, tactoids are elongated perpendicularly with respect to the flow direction and undergo relaxation."

At the beginning of the relaxation region, do the tactoids get elongated again, or do the elongated tactoids from the extension region rotate? Please clarify this.

In the revised SI, we clarified this point in the caption of the Fig. 1 as:

“Right at the beginning of the relaxation region, tactoids get elongated again but in the perpendicular direction with respect to the flow direction and undergo relaxation.”

Additionally, in the revised SI on page 7, we further explained by editing the text as following:

“Right after the elongation region, the tactoids get elongated again but in the perpendicular direction with respect to the flow direction and undergo relaxation to the equilibrium state in the relaxation region in Supplementary Figure 1. The main reason for getting elongated again of the tactoids at the beginning of the relaxation region is the high extension rate of the flow in y-direction in the expansion zone, i.e. $\dot{\epsilon}_{yy} = \partial u_y / \partial y$, compared to the shear rate in the flow direction $\dot{\Gamma}_{xy} = \partial u_x / \partial y$. For $U = 1.5 \mu\text{m s}^{-1}$ (where U is the flow speed in the straight channel before the extension zone) used in this study to understand the relaxation dynamics of the tactoids, we found the ratio of the $\dot{\epsilon}_{yy} / |\dot{\Gamma}_{xy}|$ to be always over 40 along the centerline of the channel (for details we refer the interested readers to our recent study in Supplementary Ref. 3).”

3-Fig. 4 part (b) please remove the error on the top of the graph.

We thank the Reviewer for pointing this out. In the revised manuscript, we removed the error on the top of the graph in Fig 4 part (b).